# EFFICIENT EXPERT PRUNING FOR SPARSE MIXTURE-OF-EXPERTS LANGUAGE MODELS: ENHANCING PERFORMANCE AND REDUCING INFERENCE COSTS

## ABSTRACT

The rapid advancement of large language models (LLMs) has led to architectures with billions to trillions of parameters, posing significant deployment challenges due to their substantial demands on memory, processing power, and energy consumption. Sparse Mixture-of-Experts (SMoE) architectures have emerged as a solution, activating only a subset of parameters per token, thereby achieving faster inference while maintaining performance. However, SMoE models still face limitations in broader deployment due to their large parameter counts and significant GPU memory requirements. In this work, we introduce a gradient-free evolutionary strategy named Efficient Expert Pruning (EEP) to enhance the pruning of experts in SMoE models. Specifically, EEP searches the pruning pattern and uses expert merging as a memory-efficient way of fine-tuning the pruned model. EEP relies solely on model inference (i.e., no gradient computation) and achieves greater sparsity while maintaining or even improving performance on downstream tasks. EEP can be used to reduce both the total number of experts (thus saving GPU memory) and the number of active experts (thus accelerating inference). For example, in the task-specific setting, we demonstrate that pruning up to 75% of experts in Mixtral $8 \times 7$B-Instruct results in a substantial reduction in parameters with minimal performance loss, or pruning 50% of experts and activating one fewer expert to achieve $1.41\times$ speedup. Our experiments include four different model sizes from Mixtral, Qwen1.5, and Qwen2, and utilize more than 10 datasets as well as various settings. Results show that our method outperforms the related baselines by a large margin, demonstrating a significant advancement in this direction. The results of our method can be reproduced using the code provided in the supplementary material.

## 1 INTRODUCTION

Large language models have significantly advanced, evolving into highly versatile tools (Kim et al., 2023; Brown et al., 2020; Alayrac et al., 2022; Shen et al., 2023; Zeng et al., 2023; Lu et al., 2022). As these models grow in accordance with scaling laws (Kaplan et al., 2020), the norm has shifted towards architectures with billions to trillions of parameters. However, the larger scale brings considerable deployment challenges due to increased demands on memory, processing power, and energy consumption (Zhou et al., 2024; Wan et al., 2023). In response to these challenges, there is a notable trend towards adopting sparse Mixture-of-Experts (SMoE) architectures (Shazeer et al., 2017; Fedus et al., 2022; Lepikhin et al., 2021; Hwang et al., 2023), as seen in models such as Mixtral $8 \times 7$B and $8 \times 22$B (Jiang et al., 2024), Qwen1.5-MoE-A2.7B (Bai et al., 2023), Qwen 2-57B-A14B (Qwen Team, 2024), DBRX (Team, 2023), and Grok-1 (xAI team, 2024). SMoE models activate only a subset of parameters for each token, resulting in faster inference while maintaining competitive performance compared to dense models of the same scale. For example, Mixtral $8 \times 7$B outperforms or matches Llama-2 70B (Touvron et al., 2023) and GPT-3.5 on many benchmarks, while it only activates 13B parameters to process each token. Although SMoE models have less computation per token, they remain parameter-heavy, e.g., Mixtral $8 \times 7$B has 47B parameters in total while Grok-1 reaches 314B (see Tab. 8 for other models). This limits their broader deployment due to the substantial GPU memory requirements. Additionally, their throughput may not be ideal as the

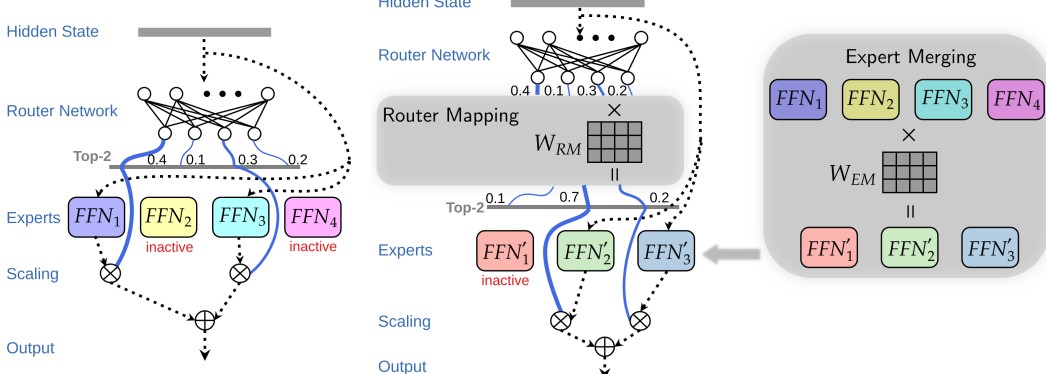

(a) A SMoE block before pruning.   (b) Parameter space designed for expert pruning and merging.

Figure 1: (a) the original SMoE block and (b) our implementation of EEP. We introduce the expert merging matrix $W_{\text{EM}}$, and the router mapping matrix $W_{\text{RM}}$, to enable the search for the optimal pruning configuration. When $W_{\text{EM}}$ and $W_{\text{RM}}$ have one-hot vectors as their rows, pruning is performed. When their elements are continuous values, routing weights and experts are aggregated to generate new weights and experts. We use an evolutionary strategy to search for the optimal $W_{\text{EM}}$ and $W_{\text{RM}}$.

batch size needs to be restricted to fit the model within the available memory. Therefore, it is vital to innovate methods that can reduce the size of SMoE models without compromising their performance.

Many studies have shown that only a subset of parameters significantly contributes to performance when applying LLMs to downstream tasks (Blalock et al., 2020; Kwon et al., 2022; Sajjad et al., 2023; Xia et al., 2022). Pruning is a crucial technique for eliminating redundancy in neural networks. It can be unstructured, achieving high sparsity while maintaining performance (Blalock et al., 2020; Frantar & Alistarh, 2023; Sun et al., 2024), or structured, removing entire channels or layers to provide computational efficiency and reduced latency (Ma et al., 2023; Tao et al., 2023; Xia et al., 2022; Hou et al., 2020; Wang et al., 2019b; Kwon et al., 2022). *One particularly efficient way is expert pruning in SMoE LLMs, a type of structured pruning with coarse granularity, which enhances overall efficiency.* Recent expert pruning methods achieve 25%-50% sparsity and accelerate inference, but struggle to maintain performance (Lu et al., 2024) or need fine-tuning, requiring substantial GPU memory and resources (Chen et al., 2022; Muzio et al., 2024). *Thus, there is a pressing need for efficient pruning methods that operate within the constraints of inference resources for SMoE LLMs.*

In this work, we propose a gradient-free evolutionary strategy to search for the optimal subset of experts or their merged models, achieving high sparsity without drastically losing performance. Our search method is divided into two phases: expert pruning and expert merging. To facilitate the search for optimal pruning configurations, we design a parameter space for router mapping and expert merging, represented by two weight matrices, $W_{RM}$ and $W_{EM}$. These matrices are applied to the router weighting and expert modules, as illustrated in Fig. 1. In the first phase, expert pruning, we search through the weight matrices to retain the most prominent experts without updating any network parameters. In the second phase, expert merging, we retrieve knowledge from the pruned experts and consolidate it into the retained experts. To these ends, $W_{RM}$ and $W_{EM}$ are set to one-hot rows in the first phase and to real numbers in the second phase. Since our method is gradient-free, it can be conducted on devices capable of inference. Our contributions can be summarized as follows:

- **Pruning the total number of experts: smaller memory consumption and better performance.** Our approach enables more aggressive pruning of experts compared to current methods (Lu et al., 2024; Muzio et al., 2024). When conduct task-specific pruning on Mixtral $8 \times 7$B-Instruct, we reduce the number of experts in each SMoE block from 8 to 2, a 72% reduction in parameters, while maintaining comparable performance across various downstream tasks. *Surprisingly, we observe that even with fewer experts EEP is still able to achiever better performance.* For instance, on the SQuAD dataset, pruning 4 out of 8 experts result in a performance increase from 53.4% to 75.4% without updating the remaining experts. We hypothesize that this is because the router network is not well-trained, which is a widely known issue for SMoE (Fedus et al., 2022; Chi et al., 2022). We also provide empirical evidence to verify this hypothesis in Sec. 5.6.

- **Pruning the number of *active* experts: better inference efficiency.** We explore the pruning of active experts and find that effective expert merging compensates for the loss of active experts

across downstream tasks. This process significantly improves efficiency without compromising the model's utility on these tasks. For instance, by reducing the active experts in Mixtral $8 \times 7B$ from two to one, we observe a prefill acceleration of up to $1.63\times$.

- **Generalization ability.** Beyond task-specific pruning, we test the performance of our method on datasets with higher diversity and out-of-distribution tasks using MMLU (Hendrycks et al., 2021). Specifically, we apply EEP on MMLU and test the resulting modeling on other out-of-distribution datasets. We observe that EEP consistently outperforms other pruning methods, demonstrating the strong generalization ability of our method.

- **A novel and efficient pruning paradigm.** Common pruning paradigm usually conducts two steps. In the first step, parameters are pruned based on predefined importance criteria (He et al., 2018; Yang et al., 2018; Ning et al., 2020). This operation often lowers performance. In the second step, retained parameters are fine-tuned through stochastic gradient descent to recover performance, which often requires substantial GPU memory and computation time. This problem also exists when downscaling the network using distillation techniques (Polino et al., 2018; Zhang et al., 2019; Aghli & Ribeiro, 2021). In contrast, EEP introduces a novel approach as a third paradigm, employing a gradient-free evolutionary strategy for both pruning and fine-tuning. Our pruned model significantly outperforms the pruned models of previous methods, while our complete processes can run on devices affordable for inference, making EEP more widely applicable.

## 2 RELATED WORK

**Sparse Mixture-of-Experts LLMs.** Shazeer et al. (Shazeer et al., 2017) introduced the sparse MoE layer, which consists of multiple experts, each being a simple feed-forward network (FFN), and a trainable router network that selects a sparse combination of the experts to process each input. Such SMoE models can significantly increase model capacity while maintaining computational efficiency. However, this utility is ideally achieved when the router accurately and evenly assigns experts to each token during training and inference. Many works focus on these challenges (Fedus et al., 2022; Lewis et al., 2021; Dai et al., 2022; Zhou et al., 2022). Recently, many SOTA LLMs adopt the SMoE structure to achieve high performance and computational efficiency simultaneously (Jiang et al., 2024; Bai et al., 2023; Team, 2023; xAI team, 2024). Additionally, Zhang et al. (Zhang et al., 2022) propose transforming non-MoE models into SMoE models to accelerate inference, and Komatsuzaki et al. (Komatsuzaki et al., 2023) upcycle pretrained models by reusing the parameters to initialize SMoE models, where all experts are replicates of the original FFNs, and then fine-tune the SMoE models.

**Pruning for LLMs.** Pruning techniques have emerged as a crucial strategy for optimizing LLMs by reducing model size and computational costs while maintaining performance. Unstructured pruning (Blalock et al., 2020; Frantar & Alistarh, 2023; Sun et al., 2024; Syed et al., 2023) entails the removal of individual weights according to specific criteria, creating sparse networks that demand specialized hardware for efficient execution. In contrast, structured pruning (Ma et al., 2023; Tao et al., 2023; Xia et al., 2022; Hou et al., 2020; Wang et al., 2019b; Kwon et al., 2022; Child et al., 2019; Xiao et al., 2024; Beltagy et al., 2020; Voita et al., 2019) eliminates entire structures, such as neurons or attention heads, facilitating more straightforward implementation on standard hardware. Within structured pruning, specific focus areas include attention mechanisms, where redundant heads are pruned to streamline the self-attention layers, and FFNs where unnecessary neurons are removed to enhance computational efficiency. Additionally, expert pruning for SMoE models selectively prunes the expert networks (Lu et al., 2024; Muzio et al., 2024; Chen et al., 2022; Koishekenov et al., 2023).

**Evolutionary Strategy for Optimization.** Evolutionary Strategies (ES) have been increasingly recognized for their robustness and flexibility in various optimization tasks, particularly where gradient-based methods fall short Wierstra et al. (2014). Notably, ES is highly effective for optimizing non-differentiable objective functions, offering a powerful alternative in scenarios where gradients are unavailable or unreliable (Salimans et al., 2017; Kharitonov, 2019; Liu et al., 2024b; Trofin et al., 2021; Liu et al., 2023). Furthermore, ES excels in discrete optimization spaces, making it suitable for a wide range of combinatorial problems (Akiba et al., 2024; Liu et al., 2024a; 2023). Recent advancements have extended the application of ES to the domain of LLMs, enabling memory-efficient fine-tuning without the need for backpropagation (Malladi et al., 2023).

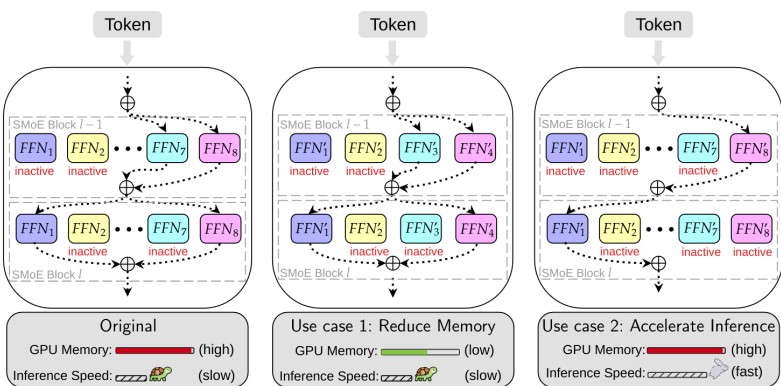

Figure 2: We leverage EEP for two purposes: reducing the total number of experts, which lowers the memory footprint (use case 1), and reducing the number of active experts, thereby accelerating inference (use case 2).

## 3 BACKGROUND OF SPARSE MIXTURE-OF-EXPERT LANGUAGE MODEL

In this section, we discuss the general concept of sparse Mixture-of-Experts (SMoE) implementation in modern decoder-only models, using the Mixtral family (Jiang et al., 2024) as an example. A schematic illustration is provided in Fig. 1a. EEP can also be applied to other types of SMoE such as Qwen (Bai et al., 2023; Qwen Team, 2024).

**Notations.** Let $X \in \mathbb{R}^{n \times d}$ represent the input to a SMoE block, where $n$ is the sequence length and $d$ is the hidden dimension. The output of the attention block is denoted by $Z \in \mathbb{R}^{n \times d}$. The main parameters in the attention block are the weight matrices for computing query, key, and value: $W_Q, W_K, W_V$. In the SMoE structure, there are $E$ experts, each represented by a feed-forward network (FFN) with parameters $\theta_i$ for the $i$-th expert. The router network, denoted by $W_R$, produces routing weights $G \in \mathbb{R}^{n \times E}$ for the sparse activation of the experts. For clarity, we omit the normalization layers and biases.

**Self-Attention Mechanism.** The self-attention mechanism computes the query, key, and value matrices as follows: $Q = XW_Q$, $K = XW_K$, $V = XW_V$. The attention scores and the output $Z$ are then computed as:

$$\text{Attention}(Q, K, V) = \text{softmax}\left(\frac{QK^\top}{\sqrt{d_k}}\right)V, \quad Z = \text{Attention}(Q, K, V)W_O, \quad (1)$$

where softmax$(\cdot)$ denotes a row-wise softmax function. The attention mechanism produces a weighted sum of the values $V$, where the weights are derived from the dot product of the queries $Q$ and keys $K$, scaled by the square root of key/query dimension $\sqrt{d_k}$. Then the weighted averaged values are mapped by the output matrix $W_O$ to $Z$.

**Router Network in SMoE Structure.** The router network $W_R$ determines which experts to activate and how to scale their outputs. The routing weights $G \in \mathbb{R}^{n \times E}$ are computed as:
$$G = \text{softmax}(ZW_R). \quad (2)$$
Sparse activation of the experts is achieved by selecting the top-$k$ routing weights for each input token. The output of the activated experts is scaled by the routing weights and aggregated to form the output of the SMoE layer $H$:[1]

$$\forall j = 1 \ldots n, \quad H_j = \sum_{i \in \text{TopK}(G_j)} G_{ji} \cdot \text{FFN}_i(Z_j), \quad (3)$$

where $\text{TopK}(G_j)$ denotes the indices of the top-$k$ routing weights for the $j$-th input token, and $\text{FFN}_i$ denotes the function of the $i$-th expert, as defined below.

**FFN as Expert.** Each expert in the SMoE structure is an independent FFN with two fully-connected layers, denoted by $W_{1i}$ and $W_{2i}$. When applying SwiGLU (Shazeer, 2020), an additional weight matrix $W_{3i}$ is introduced for the activation function. The $i$-th expert processes the input as follows:
$$\text{FFN}_i(Z_{sub}) = \text{SwiGLU}(Z_{sub}, W_{1i}, W_{3i})W_{2i}, \quad (4)$$
where $Z_{sub}$ denotes the a subset of rows in $Z$ that activates the $i$-th expert. Depending on the activation function, the parameters of the $i$-th expert are either $\theta_i = \{W_{1i}, W_{2i}\}$ or $\theta_i = \{W_{1i}, W_{2i}, W_{3i}\}$.

---

[1]The top-$k$ routing weights may be further normalized to sum to 1; this nuance is omitted here.

# 4 METHOD

In this section, we introduce our proposed approach EEP to compressing SMoE LLMs through expert pruning and merging. We parameterize the compression with two matrices and optimize them using evolutionary strategies. Our method addresses the challenges of large and complex search spaces for the pruning configuration without incurring the prohibitive computational costs associated with gradient-based optimization. The subsequent subsections elaborate on our motivation (Sec. 4.1), the configuration of the parameter space (Sec. 4.2), the evolutionary optimization strategy employed to achieve our objectives (Sec. 4.3), and the use cases we apply EEP (Sec. 4.4).

## 4.1 MOTIVATION

LLMs based on the SMoE architecture have shown remarkable performance across various NLP tasks (Jiang et al., 2024; Team, 2023; xAI team, 2024). These models leverage multiple experts, activating only a subset for any given input, thus balancing computational efficiency and model capacity. For example, 2 out of 8 experts are activated in Mixtral, striking a balance between performance and computational cost.

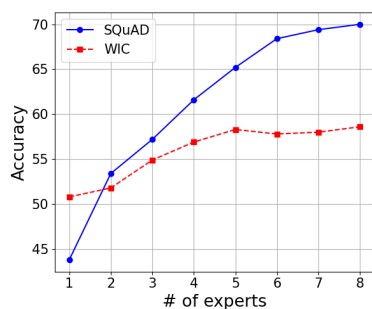

Figure 3: Performance from a single expert to an ensemble of experts.

Fig. 3 presents our investigation into the activation of different numbers of experts on Mixtral $8 \times 7$B-Instruct, revealing the following observations: **i)** Activating only a single expert does not lead to model collapse and may result in only a minimal performance drop compared to the default setting of using two experts. *This suggests that individual experts possess redundant knowledge, enabling them to maintain reasonable performance independently. This redundancy indicates potential for expert pruning.* **ii)** Conversely, activating more than 2 experts leads to a noticeable performance gain, highlighting the benefits of expert ensemble. However, the computational cost of such an ensemble is substantially higher. Wortsman et al. (Wortsman et al., 2022) have shown that merging differently fine-tuned models can efficiently substitute their ensemble, achieving similar performance with reduced computational overhead. *Therefore, expert merging can potentially strengthen a single expert, while maintaining the efficiency of inference.*

Building on these insights, we propose a two-step approach involving expert pruning followed by expert merging. Initially, we search for the optimal subset of experts given a fixed size. Subsequently, we employ expert merging to consolidate the knowledge from the pruned experts into the remaining ones. This approach not only restores the knowledge of the pruned experts but also updates the surviving experts to incorporate the collective expertise of the entire SMoE block.

## 4.2 PARAMETER SPACE FOR EXPERT PRUNING AND MERGING

**Expert Pruning and Merging Matrices.** To efficiently prune and merge experts in each SMoE block ($l = 1 \ldots L$), we introduce two key matrices: the Router Mapping matrix ($\boldsymbol{W}_{\text{RM}}^l$) and the Expert Merging matrix ($\boldsymbol{W}_{\text{EM}}^l$). For clarity, we omit the block index $l$ in this section. A schematic illustration is provided in Fig. 1b. The router mapping matrix $\boldsymbol{W}_{\text{RM}} \in \mathbb{R}^{E' \times E}$, where $E'$ is the reduced number of experts (i.e., $E > E'$), is applied to the routing weights $\boldsymbol{G}$ to reduce the dimensionality and handle fewer experts:

$$\boldsymbol{G}' = \boldsymbol{W}_{\text{RM}}\text{softmax}(\boldsymbol{Z}\boldsymbol{W}_R), \tag{5}$$

The expert merging matrix $\boldsymbol{W}_{\text{EM}} \in \mathbb{R}^{E' \times E}$ is applied to the expert weights $\{\boldsymbol{\theta}_i\}_{i=1}^{E}$ to merge $E$ experts into $E'$ experts. Each element in $\boldsymbol{W}_{\text{EM}}$ operates blockwise on the parameters of the experts. Denote $\{\omega_{j1}, \omega_{j2}, \ldots, \omega_{jE}\}$ as the $j$-th row of $\boldsymbol{W}_{\text{EM}}$ that maps the original $E$ experts to the $j$-th new expert $\boldsymbol{\theta}_j'$. We define merging as follows:

$$\boldsymbol{\theta}_j' = \{\sum_{i=1}^{E} \omega_{ji}\boldsymbol{W}_{1i}, \sum_{i=1}^{E} \omega_{ji}\boldsymbol{W}_{2i}, \sum_{i=1}^{E} \omega_{ji}\boldsymbol{W}_{3i}\}, \tag{6}$$

where the parameters of the experts are defined in Eq. (4).

**Expert Pruning Phase.** During the expert pruning phase, the low-rank matrices $\boldsymbol{W}_{\text{RM}}$ and $\boldsymbol{W}_{\text{EM}}$ are initialized with each row as a one-hot vector to ensure that only pruning occurs. Additionally, $\boldsymbol{W}_{\text{RM}}$ and $\boldsymbol{W}_{\text{EM}}$ are set as to be identical $\boldsymbol{W}_{\text{RM}} = \boldsymbol{W}_{\text{EM}}$. Consequently, these matrices only retain the

selected expert weights and their corresponding routing weights. During evolutionary search, EEP also maintains the one-hot format of $W_{\text{RM}}$ and $W_{\text{EM}}$.

**Expert Merging Phase.** In the expert merging phase, $W_{\text{RM}}$ and $W_{\text{EM}}$ are decoupled and initialized from their optimal values obtained during the pruning phase. This decoupling allows for a more flexible transformation where multiple experts can be merged, and the router weights can be updated independently. During this phase, the elements of $W_{\text{RM}}$ and $W_{\text{EM}}$ transition from discrete $0/1$ values to continuous values. This allows the matrices to perform more nuanced transformations.

**The importance of having two phases.** Theoretically speaking, only running the expert merging phase results in the same optimization space as the above two-phase approach. However, we find that having an expert pruning phase is important to give a better initialization point for the expert merging phase to achieve better results. See App. D.9 for more details.

### 4.3 Evolutionary Search for the Router Mapping and Expert Merging Matrices

The search space of the router mapping and expert merging matrices is large and complex, making it difficult to design heuristics for determining a solution, as is done in other expert pruning studies (Muzio et al., 2024; Chen et al., 2022; Lu et al., 2024). Therefore, an efficient optimization strategy is necessary. Given the substantial size of SMoE LLMs, computing gradients for optimization is computationally prohibitive for most users. As a solution, we employ a gradient-free evolutionary strategy, which has been adopted in research across other domains, such as diffusion models (Liu et al., 2023; 2024b). **Our algorithm is detailed in Alg. 1**. The key points are summarized as follows:

**i)** Initially, we populate the search space using random initialization. **ii)** During the evolutionary search, each set of router mapping and expert merging matrices is treated as an individual. In each iteration, only the top-performing individuals are selected as parents to produce the next generation through crossover and mutation. Specifically, during crossover, we randomly combine the entries of the matrices from two parents or select one parent's matrices entirely. For mutation, we introduce random Gaussian noise to the matrices, ensuring stochastic variations. This process conserves beneficial adaptations while discarding detrimental modifications, enhancing the optimization process. **iii)** This evolutionary reproduction process is repeated for a predetermined number of iterations within each search phase, updating the population with newly generated individuals. **iv)** Upon completion of the search process, the best individual is selected as the output of our search algorithm.

### 4.4 Use Cases

We explore two applications of EEP: expert pruning and expert activation pruning. **In expert pruning**, EEP searches for optimal router mapping ($W_{\text{RM}}$) and expert merging matrices ($W_{\text{EM}}$) to minimize the *total number of experts* while maintaining high performance. **For expert activation pruning**, the goal is to achieve strong performance with *only one active expert per token*. Here, we use the same EEP search algorithm to conduct expert and router networks optimization by updating the $W_{\text{RM}}$ and $W_{\text{EM}}$ matrices, while only activates one expert during inference. Fig. 2 illustrates these two use cases. *Additionally, we investigate the combination of these two approaches, reducing both the total number of experts and the number of active experts simultaneously (see Sec. 5.3).*

## 5 Experiments

In this section, we validate the effectiveness of our method by considering use cases mentioned in Sec. 4.4. In Sec. 5.1, we introduce the experimental settings. In Sec. 5.2, we investigate the first use case, expert pruning, by applying EEP to reduce the total number of experts. In Sec. 5.3, we further explore expert activation pruning. We also examine a composite case where both the total number of experts and the number of active experts are reduced. In Sec. 5.4, we present the experimental results on larger and more diverse datasets, as well as performance on out-of-distribution datasets. In Sec. 5.5, we profile memory usage and inference speed to demonstrate that our method achieves significant improvements compared to the full SMoE models. In Sec. 5.6 we provide insights on the observation of fewer experts but higher performance. More results, including experiments on larger datasets and other models, and running time of EEP, can be found in App. D.

### 5.1 Experimental Settings

Our main results are based on the popular SMoE models Mixtral $8\times7$B (Jiang et al., 2024). We also include a larger model, Mixtral $8\times22$B (Jiang et al., 2024) and another types of SMoE model Qwen1.5-A2.7B (Bai et al., 2023) and Qwen2-A14B (Qwen Team, 2024) to demonstrate the gen-

Table 1: Results of expert pruning on Mixtral $8\times7$B-Instruct on SuperGLUE. **Bold** values indicate the best across all methods; underlined values show the best without parameter updates (i.e., excluding EEP (Prune+Merge)).

| Expert | Method | COPA | MultiRC | WIC | WSC | RTE | BoolQ | CB | ReCoRD | DROP | SQuAD | Avg. |
|---|---|---|---|---|---|---|---|---|---|---|---|---|
| Num=8 | Full Model | 89.0 | 83.0 | 51.8 | 63.5 | 73.2 | 77.4 | 51.7 | 50.3 | 30.6 | 53.4 | 62.4 |
| Num=4 | Random | 63.8 | 49.4 | 37.6 | 43.3 | 45.1 | 50.2 | 38.7 | 35.1 | 27.4 | 58.3 | 44.9 |
| | Frequency (Muzio et al., 2024) | 63.0 | 74.8 | 36.0 | 34.6 | 18.1 | 71.0 | 30.4 | 41.6 | 29.9 | 58.2 | 45.8 |
| | Soft Activation (Muzio et al., 2024) | 73.0 | 30.6 | 51.4 | 37.5 | 41.9 | 40.4 | 17.9 | 36.8 | 33.3 | 10.2 | 37.3 |
| | NAEE (Lu et al., 2024) | 87.0 | 76.0 | 52.6 | 64.5 | 61.7 | 77.2 | 51.7 | 50.4 | 30.6 | 53.0 | 60.5 |
| | EEP (Prune Only) | 95.0 | 81.2 | 57.8 | 67.3 | 74.0 | 82.8 | 69.6 | 60.0 | 37.3 | 75.2 | 70.3 |
| | EEP (Prune+Merge) | **99.0** | **84.6** | **65.0** | **73.1** | **76.9** | **84.8** | **75.0** | **63.6** | **39.7** | **80.6** | **74.2** |
| Num=2 | Random | 36.8 | 22.3 | 13.6 | 15.0 | 28.4 | 15.5 | 38.6 | 16.9 | 18.3 | 36.9 | 24.2 |
| | Frequency (Muzio et al., 2024) | 51.0 | 17.6 | 8.8 | 1.9 | 48.4 | 30.6 | 35.7 | 10.4 | 14.9 | 9.2 | 24.9 |
| | Soft Activation (Muzio et al., 2024) | 33.0 | 18.2 | 49.4 | 18.5 | 15.2 | 1.8 | 32.1 | 4.4 | 11.7 | 50.0 | 23.4 |
| | NAEE (Lu et al., 2024) | 75.0 | 42.4 | 48.4 | 49.0 | 54.5 | 49.8 | 19.6 | 42.0 | 31.2 | 58.2 | 47.0 |
| | EEP (Prune Only) | 76.0 | 63.8 | 51.8 | 63.5 | 64.3 | 70.6 | 58.9 | 47.2 | 37.1 | 64.0 | 59.7 |
| | EEP (Prune+Merge) | **93.0** | **71.6** | **58.6** | **65.4** | **69.0** | **75.6** | **66.1** | **47.2** | **38.4** | **70.2** | **65.6** |

eralization of our methods. We use the instruction fine-tuned version of these models. We use datasets from the SuperGLUE (Wang et al., 2019a) and many other datasets for the generation tasks, including SQuAD (Rajpurkar et al., 2016), DROP (Dua et al., 2019), ARC (Clark et al., 2018), HellaSwag (Zellers et al., 2019), OBQA (Mihaylov et al., 2018), WinoGrande (Sakaguchi et al., 2021), and GSM8K (Cobbe et al., 2021). For each individual dataset, we randomly sample a subset of data as the training set to conduct evolutionary search (for EEP) or importance estimation (for baselines), and use the test set (or the rest of the dataset) for evaluation. We evaluate EEP and baselines under both task-specific setting (training set and test set are identically distributed) and global setting (non-identically distributed). Additional details can be found in App. A.

**Evaluation.** We mainly adopt a generation-based evaluation approach for all datasets, while the results of log-probability-based evaluation can be found in App. D.2. Specifically, we use the instruction fine-tuned model to generate answers directly in response to the given questions and apply template matching to determine the correctness of the answers. Our evaluation protocol primarily follows the implementation of OpenCompass (Contributors, 2023) for the design of question prompts, types of templates, and matching criteria, with a few modifications to better suit the Mixtral family of models. Examples of prompts and model outputs can be found in App. E and App. F.

**Baselines.** Since our method aims to compress the instruction fine-tuned SMoE models on downstream tasks, we consider the zero-shot performance as our main baseline to show that EEP can achieve a significant decrease on the memory footprint and/or computation overhead during the inference time while barely loss or even achieve better performance. Additionally, we compare EEP with four other types of baseline to demonstrate the effectiveness of the designed search space and the evolutionary-search-based tuning method: **(1)** `Random` selection of pruned experts, **(2&3)** Pruning the experts with the lowest `frequency` of being activated or the lowest `soft` activation values (Muzio et al., 2024), and **(4)** `NAEE` (Lu et al., 2024), which exhaustively evaluates the loss between the full model and all pruning choices for each layer and select the one with the lowest loss. For the use case of decreasing the active number of experts, we select the dynamic skipping method proposed by `NAEE` (Lu et al., 2024) as an additional baseline. More details are given in App. A.

## 5.2 REDUCING THE TOTAL NUMBER OF EXPERTS

We apply EEP to search for the optimal pruning configuration, parameterized by the router mapping matrix $W_{\text{RM}}$ and the expert merging matrix $W_{\text{EM}}$, for maintaining 4 experts and 2 experts. In this experiment we consider task-specific pruning, i.e. a small set of training data is available for search (EEP) or importance estimation (baselines). EEP (Prune Only) indicates the results from solely conducting the expert pruning phase as described in Sec. 4.2. In contrast, EEP (Prune + Merge) shows the results after the complete evolutionary search process. The results are shown in Tab. 1, and we discuss them below. `Random` is conducted 30 times, and we present the mean results here, deferring the complete results to App. D.7.

**EEP exploits expert-wise redundancy on downstream tasks**. Based on the results obtained from the pruning phase of EEP, retaining only 4 experts allows the model to achieve better performance and lower computational costs simultaneously on most datasets, except for MultiRC. Even with a particularly low budget of retaining only 2 experts, EEP can still achieve comparable or even better performance than the full model on five datasets, with some datasets showing significant improvements over the best baseline (e.g., 58.9 vs. 51.7 on CB and 64.0 vs. 53.4 on SQuAD). For the remaining datasets, model collapse is avoided.

Table 2: Additional results of expert pruning on Mixtral 8×7B-Instruct.

| Expert | Method | ARC-c | ARC-e | HellaSwag | OBQA | WinoGrande | gsm8k | Avg. |
|---|---|---|---|---|---|---|---|---|
| Num=8 | Full Model | 85.8 | 91.7 | 71.5 | 89.4 | 60.5 | 61.0 | 76.6 |
| Num=6 | NAEE | 80.0 | 89.6 | 68.3 | 88.0 | 57.3 | 55.0 | 73.0 |
| | EEP (Prune+Merge) | **84.4** | **91.7** | **71.4** | **91.2** | **64.2** | **66.0** | **78.2** |
| Num=4 | NAEE | 76.2 | 85.4 | 65.1 | 85.6 | 53.9 | 41.0 | 67.9 |
| | EEP (Prune+Merge) | **80.7** | **87.1** | **67.3** | **88.3** | **61.1** | **54.0** | **73.1** |

Table 3: Results of expert pruning on Mixtral 8 × 22B-Instruct. **Bold** values indicate the best across all methods; underlined values show the best without parameter updates (i.e., excluding EEP (Prune+Merge)).

| Budget | Method | WIC | WSC | BoolQ | CB | SQuAD | Avg. |
|---|---|---|---|---|---|---|---|
| Num=8 | Full Model | 68.2 | 81.7 | 90.2 | 46.5 | 45.8 | 66.5 |
| Num=4 | Random | 27.0 | 30.2 | 37.8 | 34.6 | 37.2 | 33.4 |
| | Frequency (Muzio et al., 2024) | 0.0 | 38.5 | 76.6 | 57.1 | 50.6 | 30.6 |
| | Soft Activation (Muzio et al., 2024) | 25.2 | 60.6 | 6.4 | 60.7 | 54.2 | 41.4 |
| | NAEE (Lu et al., 2024) | 64.0 | 68.3 | 78.4 | 33.9 | 52.4 | 59.4 |
| | EEP (Prune Only) | 70.2 | 84.2 | **89.6** | 75.0 | 71.4 | 78.1 |
| | EEP (Prune+Merge) | **72.2** | **87.5** | 89.6 | **78.6** | **74.0** | **80.4** |
| Num=2 | Random | 13.9 | 10.1 | 11.0 | 24.9 | 15.6 | 15.1 |
| | Frequency (Muzio et al., 2024) | 0.0 | 0.0 | 0.0 | 0.0 | 0.0 | 0.0 |
| | Soft Activation (Muzio et al., 2024) | 2.4 | 1.9 | 3.6 | 19.6 | 52.6 | 16.0 |
| | NAEE (Lu et al., 2024) | 34.0 | 32.7 | 45.0 | 16.1 | 50.0 | 30.6 |
| | EEP (Prune Only) | 57.8 | 63.5 | 76.0 | 50.0 | 71.0 | 63.7 |
| | EEP (Prune+Merge) | **59.6** | **65.4** | **76.4** | **58.9** | **75.0** | **67.1** |

**EEP is more effective than other baseline methods for selecting pruned experts.** Comparing the results of other methods, we find that EEP is more effective for identifying the optimal pruning pattern. Random sampling of experts results in low mean accuracy and high variance. Pruning experts based on selection frequency also performs poorly on most datasets and has a high probability of collapse under high sparsity. `NAEE` can nearly maintain the performance of the full model when retaining four experts. However, EEP surpasses all methods by a large margin across all datasets.

**Expert merging brings significant improvements after pruning**. As shown in the last row for each pruning rate in Tab. 1, the results after expert merging exceed those obtained through the expert pruning phase alone. Specifically, expert merging achieves a general improvement on almost all datasets. On WIC, CB, and SQuAD under both pruning rates, and on WSC when four experts are retained, the accuracy improvement reaches 5%~7%, demonstrating its effectiveness in restoring the knowledge of pruned experts and enhancing individual experts. Additionally, we find expert merging to be an effective method for fine-tuning SMoE LLMs (i.e., keeping the number of total and active experts); the results of this are presented in Tab. 17.

**Generality across models.** With the promising results of Mixtral 8×7B-Instruct model, we further apply EEP to a larger model: Mixtral 8×22B-Instruct (Jiang et al., 2024), Qwen1.5-MoE-A2.7B-Chat (Bai et al., 2023), and Qwen2-MoE-A14B-Chat (Qwen Team, 2024). We conduct experiments on fewer datasets due to the constraint of computational resource. Results are shown at Tab. 3, Tab. 9, and Tab. 10, respectively. EEP also achieves a strong improvement and above observations are still held, which indicates the scaling-up (down) ability of EEP towards large (small) SMoE models.

**More Tasks.** We further compare EEP with the strongest baseline, `NAEE` (Lu et al., 2024) on several additional datasets to better demonstrate the effectiveness of EEP, as shown in Tab. 2 and Tab. 11. EEP consistently outperforms the strongest baseline `NAEE`. Notably, it also achieves performance that is comparable to or even better than the full model on many datasets.

### 5.3 REDUCING THE NUMBER OF ACTIVE EXPERTS

Next, we present the experimental results for the second use case: decreasing the number of active experts. We modify the number of active experts by changing the top-k from $k = 2$ to 1 while applying EEP to restore model performance. In this experiment we also consider the task-specific setting. We evaluate our method with two different total numbers of experts (8 and 4). The results are presented in Tab. 4. We summarize the observations below.

**EEP can improve individual experts through expert merging, allowing a single expert to handle the inference.** Keeping the total number of experts at 8 and reducing the number of active experts to

Table 4: Results of active expert pruning on Mixtral $8 \times 7$B. Bold values show the best performance. "Active" indicates the average number of experts active per token. Avg. stands for average.

| Total | Active | Method | WIC | WSC | BoolQ | CB | SQuAD | Avg. |
|---|---|---|---|---|---|---|---|---|
| 8 | 2 | Full Model | 51.8 | 63.5 | 77.4 | 51.7 | 53.4 | 59.6 |
| | 1
1.4~1.5 | Full Model
Dyn (Lu et al., 2024) | 50.8
50.0 | 48.1
59.6 | 66.0
72.8 | 48.2
46.4 | 43.8
44.8 | 51.4
54.7 |
| | 1 | EEP | **59.2** | **70.2** | **79.0** | **66.1** | **51.8** | **65.3** |
| 4 | 1
1.4~1.5 | NAEE (Lu et al., 2024)
NAEE+Dyn (Lu et al., 2024) | 48.6
43.4 | 20.2
61.5 | 56.2
36.2 | 33.9
53.6 | 51.8
53.4 | 42.1
49.6 |
| | 1 | EEP | **55.8** | **70.2** | **74.4** | **64.3** | **72.0** | **67.3** |

Table 5: Results using generation-based evaluation under general OOD task pruning setting. MMLU-val stands for the 7 dataset in MMLU used for validation, which is excluded from average calculation.

| Expert | Method | MMLU-val | WIC | WSC | RTE | BoolQ | CB | DROP | SQuAD | Avg. |
|---|---|---|---|---|---|---|---|---|---|---|
| Num=8 | Full Model | 72.6 | 51.8 | 63.5 | 73.3 | 77.4 | 51.8 | 32.0 | 52.8 | 57.5 |
| Num=6 | NAEE
EEP (Prune+Merge) | 69.4
71.4 | 54.2
52.4 | 60.6
69.2 | 55.2
52.0 | 69.4
83.2 | 53.6
44.6 | 30.7
34.4 | 45.2
65.2 | 52.7
**57.3** |
| Num=4 | NAEE
EEP (Prune+Merge) | 63.6
64.6 | 55.8
55.0 | 65.4
60.6 | 54.9
70.8 | 76.2
82.8 | 33.9
51.8 | 29.6
34.1 | 55.0
58.2 | 53.0
**59.0** |

1 consistently leads to a decline in baseline performance. However, by optimizing the model with EEP, we introduce a reliable improvement that mitigates this gap, resulting in comparable or even better performance than the full model. Note that when the total number of experts is maintained, there is no expert pruning phase; only expert merging is applied for EEP.

**The two use cases can be combined through EEP.** By retaining fewer experts while reducing the number of active experts, we achieve significant savings *in both GPU memory and inference time* (see Sec. 5.5). EEP can be directly applied in this scenario. Results show that with 4 total experts and 1 active expert, EEP achieves performance comparable to or even better than the full model.

## 5.4 IN-DISTRIBUTION AND OUT-OF-DISTRIBUTION GENERALIZATION ON DIVERSE DATASETS

Next, we evaluate EEP and the baselines in a global setting, where the downstream task is complex rather than specific, and the test data may diverge from the data used for pruning. Specifically, we conduct EEP on a larger dataset, MMLU. We randomly split all 57 datasets in MMLU into two subsets containing 50 and 7 datasets, as the training dataset and the validation dataset, respectively. EEP is conducted on the training dataset and the validation dataset is used to select searched patterns. Then we test the searched results on the validation dataset and other out-of-distribution (OOD) datasets. Results shown in Tab. 5, Tab. 12 and Tab. 13, demonstrate that EEP outperforms baseline methods on both the MMLU validation dataset and the OOD dataset. This indicates that EEP has the ability to handle large and diverse datasets and exhibits a certain level of generalization capability.

## 5.5 IMPROVEMENTS IN MEMORY USAGE AND INFERENCE SPEED

We profile the memory overhead and inference speed of Mixtral $8 \times 7$B model for the two use cases. We conduct tests on SQuAD with a batch size of 256 using two NVIDIA A100 GPU cards. We report the peak memory usage and the wall-time acceleration ratio in Tab. 6. As shown in Tab. 6, retaining only 4 and 2 experts from the whole model decreases the memory overhead by 47% and 71%, respectively. Additionally, reducing the total number of experts improves inference speed due to higher parallelism, achieving a speedup of $1.11\times$ and $1.18\times$ with 4 and 2 experts, respectively. In the use case of reducing active experts, an acceleration ratio of $1.24\times$ is achieved. Finally, when combining the two use cases with 4 total experts and 1 active expert per token, EEP saves 47% of GPU memory and achieves a $1.41\times$ increase in inference speed. The profiling results indicate that EEP can significantly reduce the memory consumption and computational cost of SMoE LLMs.

## 5.6 WHY FEWER EXPERTS LEADS TO BETTER PERFORMANCE

An intriguing phenomenon in our experiment is that reducing the number of experts can improve performance, as shown in Tabs. 1 and 3. Notably, this happens without fine-tuning the remaining parameters. Typically, the router network is implemented as a smaller network, such as a one-layer perceptron. This makes it challenging to accurately partition the high-dimensional hidden space among experts. The issue of imbalanced activation has been identified in several works (Fedus et al., 2022; Chi et al., 2022). To illustrate this issue more clearly, we conduct a simple experiment: we find

Table 6: Profiling the memory footprint and inference speedup of Mixtral $8 \times 7$B.

| Total | Active | Method | Speedup | GPU Mem(GB) |
|---|---|---|---|---|
| 8 | 2 | Full Model | $1.0\times$ | 88.6 |
| | 1 | EEP | $1.24\times$ | |
| 4 | 2 | EEP | $1.11\times$ | 46.6 |
| | 1 | EEP | $1.41\times$ | |
| 2 | 2 | EEP | $1.18\times$ | 25.6 |

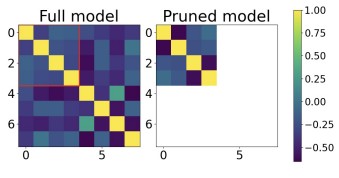

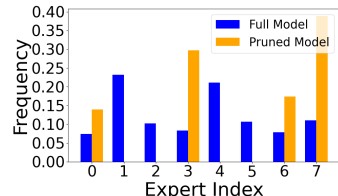

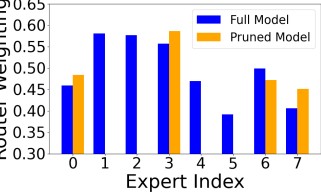

(a) Activation (1/0 means activated/not activated) correlation across experts before and after pruning.

(b) Accumulated activation times before and after pruning.

(c) Accumulated routing weights before and after pruning.

Figure 4: Statistics of the expert activation patterns before and after the Expert Pruning Phase. The data represents the first transformer block of Mixtral $8 \times 7$B-Instruct on the SQuAD dataset. In (a), four retained experts are re-indexed from 0 to 3 for clarity.

Table 7: Results of using the 3rd and 4th experts in the layers indexed by 12,16,20,24,28 of Mixtral $8 \times 7$B.

| Modified Layers index | None(The Original Model) | {12,24} | {12,16,20} | {12,16,20,24} | {12,16,20,24,28} |
|---|---|---|---|---|---|
| Accuracy on SQuAD | 53.4 | 62.0 | 67.0 | **67.4** | **67.4** |

that activating lower-ranked experts can improve performance. Specifically, we let several layers of the model select experts with 3rd and 4th rank, as given by the router, rather than 1st and 2nd rank. If the router networks are well-trained, choosing lower-ranked experts would typically degrade the model. However, we are able to find some layers improving the test performance by forcing them to choose lower-ranked experts on the SQuAD dataset, as shown in Tab. 7.

Since the router network does not function optimally before pruning, we hypothesize that there may be potential for improvement by enabling the router to focus on a smaller subset of experts. Empirical evidence potentially supporting our hypothesis is that the router network operates differently after expert pruning. This change occurs because the pruning process eliminates some experts, and the routing weights for the rest experts are normalized to sum to one. As shown in Fig. 4, we observe distinct patterns in the accumulated activation times of the experts, their accumulated routing weights, and the activation correlation across experts. Further demonstration can be found in App. D.10.

Another potential reason is that EEP optimizes the use case metrics directly. This is possible as *EEP does not require gradient computation and can be applied on top of any metric.* In contrast, the pertaining requires differentiable loss and therefore has to use losses such as likelihood which is not directly related to the metrics users care about.

## 6 CONCLUSION

In this work, we present EEP, a gradient-free evolutionary search method optimized for pruning within an efficient parameter space. Through extensive experiments on various datasets and settings, we demonstrate that EEP achieves superior performance and greater sparsity compared to baseline methods. Additionally, we make a novel observation that the performance of SMoE models on downstream tasks can be improved under task-specific pruning, even without fine-tuning the remaining parameters. We discuss the potential reasons for this phenomenon, suggesting that a good pruning configuration may lead to a more effective routing mechanism by reducing the complexity the router network needs to manage.

**Limitations.** EEP can run on a device that is solely capable of inference. However, if the dataset used for the search is too large, the search process can require a long computation time. We leave the optimization of search speed to future work.

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

## A ADDITIONAL DETAILS ON EXPERIMENTAL SETTINGS

### A.1 EEP CONFIGURATIONS

**Search Space.** As mentioned in Sec. 5, to avoid optimizing too many parameters, we split the weights of all experts into several groups. The merging coefficients $W_{EM}$ and $W_{RM}$ within the same group are shared. Most of our main results are obtained by uniformly splitting all weights into four groups based on their depth, except for the experiments on the RTE, ReCoR, and DROP datasets in Tab. 1. We find that for these datasets, setting each layer as an independent group performs significantly better than using only four groups during the pruning phase. More detailed results can be found in App. D.8. For other datasets, we maintain the current setting without exploring other configurations, as it consistently yields good performance.

**Search Process.** We apply a two-stage search method as discussed in Sec. 4.2. The pruning phase consists of 40 iterations, followed by 160 iterations for the expert merging phase. At each iteration, we evaluate the accuracy on the training set and use this metric as the score for all individuals of merging coefficients in the population. Examples of the performance curve over the search iterations are provided in App. D.8.

**Selected Datasets for OOD Evaluation.** In Sec. 5.4, we randomly select 7 datasets for OOD test. These datasets are: **(1)** *lukaemon_mmlu_electrical_engineering*, **(2)** *lukaemon_mmlu_professional_accounting*, **(3)** *lukaemon_mmlu_high_school_macroeconomics*, **(4)** *lukaemon_mmlu_high_school_computer_science*, **(5)** *lukaemon_mmlu_business_ethics*, **(6)** *lukaemon_mmlu_miscellaneous*, and **(7)** *lukaemon_mmlu_high_school_psychology*.

### A.2 BASELINE CONFIGURATIONS

To evaluate the effectiveness of reducing the total number of experts, we compare our method against four baseline approaches: **(1)** `Random` selection of pruned experts, **(2)** pruning experts with the

lowest `frequency` of activation, **(3)** pruning experts with the lowest `soft activation` values, and **(4)** `NAEE` Lu et al. (2024), which exhaustively evaluates the discrepancy between the full model and all pruning choices for each layer and selects the one with the lowest discrepancy. For reducing the number of active experts, we adopt the dynamic skipping scheme from `NAEE` as a baseline approach.

For random selection, we uniformly sample a corresponding number of experts from all 8 experts in each layer. The full results with error margins for random selection are presented in Tab. 19.

For the frequency-based method, we run the model on the training set and count the number of times each expert is activated. We then prune the experts with the lowest frequency in each layer.

For the soft activation method, we run the model on the training set and accumulate the router weighting (soft activation value) for each expert. We then prune the experts with the lowest accumulated values in each layer.

For `NAEE`, we enumerate all pruning choices for each layer and select the one with the smallest output discrepancy compared to the full model. We use a batch of calibration data with a size of 64 to calculate the discrepancy. For the dynamic skipping scheme, we run the model on the entire training set to determine the median value of the ratio between the two largest routing weights for each layer. During validation, we dynamically skip the expert with the second-largest routing weight if the ratio between its weight and the largest weight is below the threshold. This results in an average of approximately 1.5 active experts.

## B  SIZE OF CURRENT SMoE LLMs

Tab. 8 shows the basic parameter information of modern SMoE Large LLMs.

Table 8: Active Parameters, Total Parameters, and Parameters of the Experts for Various Models

| Model | Active Parameters | Total Parameters | Parameters of Experts |
| --- | --- | --- | --- |
| Mixtral 8x7B | 13B | 47B | 45B |
| Mixtral 8x22B | 39B | 141B | 136B |
| Grok-1 | 79B | 314B | 313B |
| DBRX | 36B | 132B | 128B |
| Qwen 1.5-MoE-A2.7B | 2.7B | 14.3B | 13.2B |
| Qwen 2-57B-A14B | 14B | 57B | 49B |

## C  ALGORITHM DETAILS

Alg. 1 presents the details of EEP. The notations are consistent with those in Sec. 4.2. For the Crossover operation, we combine the merging coefficients of the parent models along the dimension of the retained experts. For the Mutate operation, we perturb the merging coefficients. Specifically, during the pruning phase, we randomly replace the pruned experts with other experts and set the router weights accordingly. In the expert merging phase, we perturb the merging coefficients element-wise by adding Gaussian noise.

## D  ADDITIONAL RESULTS

### D.1  RESULTS WITH OTHER MODELS

In this section, we further apply EEP to the Qwen 1.5 Bai et al. (2023) and Qwen 2 Qwen Team (2024) SMoE models. Results can be found in Tab. 9 and Tab. 10. The same observations in Sec. 5 hold for these models: **(1)** EEP selects better pruning patterns than other baseline methods without updating the remaining parameters, and **(2)** expert merging brings improvements in most cases.

For the Qwen1.5-MoE-A2.7B-Chat Bai et al. (2023), we notice that other methods are prone to collapse. Conversely, the situation is the opposite for the Qwen2-MoE-A14B-Chat model Qwen Team (2024). Most baseline methods can maintain the performance of the full model with an extremely

---

**Algorithm 1** Evolutionary Search of EEP

---

**Require:**

$\Theta = \{\boldsymbol{\theta}_1^l, \boldsymbol{\theta}_2^l, \cdots, \boldsymbol{\theta}_E^l\}_{l=1}^L$: Full set of expert weights across all $L$ SMoE blocks.

$\mathcal{F}$: The metric evaluator.

**Symbols:**

$P$: The whole *P*opulation of matrix configurations.

$CP$: The *C*andidate *P*arents set of each loop, from which a parent configuration is selected.

$NG$: The *N*ext *G*eneration newly mutated from the parent configurations in each loop.

$\boldsymbol{W} = \{\boldsymbol{W}_{\text{EM}}^l, \boldsymbol{W}_{\text{RM}}^l\}_{l=1}^L$: Full set of the search parameters across all $L$ SMoE blocks.

**Hyperparameters:**

**Epoch**: Number of loops for the entire search process.

$\mathbf{M}_{CP}$: Maximum size of the candidate parents set $CP$.

**Iter**: Maximum number of mutations in each loop.

**Search Process:**

1: $P \leftarrow \varnothing$
2: Initialize a set of random matrices $\boldsymbol{W}_{\text{init}}$, ensuring that each row is a one-hot vector.
3: $P \leftarrow P \cup \{(\boldsymbol{W}_{init}, \mathcal{F}(\boldsymbol{W}_{init}))\}$
4: **for** $r = $ *Expert Pruning Phase*, *Expert Matching Phase* **do**
5:     **for** $t = 1, \cdots, Iters$ **do**
6:         $NG \leftarrow \varnothing$
7:         **for** $i = 1, \cdots, Epochs$ **do**
8:             $CP \leftarrow \{\boldsymbol{W}_i | \mathcal{F}(\boldsymbol{W}_i \cdot \Theta) \text{ ranks within the top } min(\mathbf{M}_{CP}, |P|) \text{ in } P\}$
9:             $\boldsymbol{W}_f, \boldsymbol{W}_m \xleftarrow{\text{Random Sample}} CP$
10:           $\boldsymbol{W}_{new} \leftarrow \text{Mutate}(\text{Crossover}(\boldsymbol{W}_f, \boldsymbol{W}_m))$
11:           $NG \leftarrow NG \cup \{(\boldsymbol{W}_{new}, \mathcal{F}(\boldsymbol{W}_{new}))\}$
12:         **end for**
13:         $P \leftarrow P \cup NG$
14:     **end for**
15: **end for**
16: $\boldsymbol{W}^* \leftarrow \underset{\boldsymbol{W} \in P}{\arg\min} \, \mathcal{F}(\boldsymbol{W})$
17: **return** $\boldsymbol{W}^*$

---

low number of experts retained. In face, we observe that the experts in the Qwen2-MoE-A14B-Chat model are specifically homogeneous, as the model's performance is largely maintained even when only one random expert is activated per token. However, according to the information provided in their technical report, both Qwen1.5-MoE-A2.7B andQwen2-MoE-A14B employ upcycling and 64 experts per layer. We thus speculate that other training configurations, such as sizes and optimizer hyperparameters, lead to different final statuses. Nevertheless, EEP always achieves comparable or better performance than the full model and outperforms all baseline methods across settings, demonstrating its adaptability to different SMoE models.

## D.2 RESULTS USING LOG-PROBABILITY-BASED EVALUATION

In addition to the generation-based evaluation method used in Sec. 5, log-probability is another widely adopted evaluation method for LLMs. In this section, we present the performance of EEP using log-probability-based evaluation. We begin by applying this evaluation method during the search phase, with the results reported in Tab. 11. Additionally, we use the general pruning setting (i.e., pruning on MMLU dataset and testing the results on down-stream tasks using log-probability-based evaluation), with the results are shown in Tab. 12 and Tab. 13. EEP achieves consistent better performance than baselines and outperform the full model under many settings.

## D.3 MORE RESULTS ON OOD GENERALIZATION TEST

We evaluate the searched matrices from MMLU on additional downstream datasets. The results are shown in Tab. 13. The findings show that EEP outperforms the baseline method in most cases, indicating the effectiveness of EEP.

Table 9: Results of expert pruning on Qwen1.5-MoE-A2.7B-Chat. Bold values indicate the best performance; underlined values show the best without updating remaining parameters. For NAEE, due to the excessive number of combinatorial possibilities, we only randomly select 5k of them for each layer.

| Budget | Method | WIC | WSC | BoolQ | CB | SQuAD | Avg. |
|---|---|---|---|---|---|---|---|
| Num=60 | Full Model | 51.4 | 46.2 | 73.6 | 32.1 | 68.6 | 54.4 |
| Num=30 | Random | 3.7±12.1 | 7.6±14.3 | 8.1±12.9 | 5.6±8.4 | 19.5±23.0 | 8.9 |
| | Frequency | 55.6 | 9.6 | 2.4 | 0.0 | 17.9 | 21.7 |
| | Soft Activation | 51.4 | 30.8 | 0.4 | 44.6 | 28.0 | 31.0 |
| | NAEE | 0.0 | 0.0 | 1.6 | 0.0 | 34.6 | 7.2 |
| | EEP (Prune Only) | 59.8 | 59.6 | **78.0** | 71.4 | 70.6 | 67.9 |
| | EEP (Prune+Merge) | **62.6** | **66.3** | **81.4** | **76.9** | **71.4** | **71.7** |
| Num=15 | Random | 1.4±5.9 | 0.5±1.3 | 2.0±4.1 | 4.3±10.6 | 1.1±3.4 | 1.9 |
| | Frequency | 0.0 | 0.0 | 7.8 | 16.1 | 0.0 | 4.9 |
| | Soft Activation | 26.2 | 3.9 | 0.0 | 0.0 | 25.4 | 11.1 |
| | NAEE | 0.0 | 1.0 | 5.2 | 0.0 | 0.0 | 1.2 |
| | EEP (Prune Only) | 51.0 | 36.5 | 45.4 | 60.7 | 57.6 | 50.2 |
| | EEP (Prune+Merge) | **54.4** | **63.5** | **58.2** | **76.9** | **60.8** | **62.7** |

Table 10: Results of expert pruning on Qwen2-MoE-A14B-Chat. Bold values indicate the best performance; underlined values show the best without updating remaining parameters. For NAEE, due to the excessive number of pruning patterns, we only randomly select 2k of them for each layer.

| Budget | Method | WIC | WSC | BoolQ | CB | SQuAD | Avg. |
|---|---|---|---|---|---|---|---|
| Num=64 | Full Model | 60.2 | 68.3 | 88.8 | 67.9 | 74.4 | 71.9 |
| Num=8 | Random | 55.3±7.1 | 61.6±5.6 | 78.7±7.3 | 35.4±17.6 | 79.7±2.4 | 62.1 |
| | Frequency | 58.8 | 59.6 | 79.4 | 46.4 | 78.2 | 64.5 |
| | Soft Activation | 60.8 | 64.4 | 82.6 | 14.3 | 75.2 | 59.5 |
| | NAEE | 56.6 | 60.6 | 82.6 | 41.1 | 81.2 | 64.4 |
| | EEP (Prune Only) | 61.8 | 72.1 | **85.8** | 76.8 | 85.6 | 76.4 |
| | EEP (Prune+Merge) | **63.4** | **75.0** | **85.8** | **85.7** | **87.0** | **79.4** |
| Num=4 | Random | 56.5±1.9 | 59.8±5.2 | 79.1±4.0 | 32.1±15.0 | 78.0±2.4 | 61.1 |
| | Frequency | 56.8 | 60.6 | 83.2 | 17.9 | 80.0 | 59.7 |
| | Soft Activation | 59.2 | 61.5 | 81.6 | 17.9 | 77.6 | 59.6 |
| | NAEE | 55.0 | 61.5 | 75.8 | 21.4 | 79.6 | 58.7 |
| | EEP (Prune Only) | 62.0 | 65.4 | 84.6 | 69.6 | 80.6 | 72.4 |
| | EEP (Prune+Merge) | **63.8** | **72.1** | **85.8** | **80.4** | **84.2** | **77.3** |
| Num=2 | Random | 56.4±1.4 | 58.2±3.7 | 77.8±4.5 | 26.5±9.6 | 76.4±1.9 | 59.1 |
| | Frequency | 58.0 | 60.6 | 79.6 | 42.9 | 72.4 | 62.7 |
| | Soft Activation | 57.4 | 65.4 | 71.4 | 62.5 | 76.8 | 66.7 |
| | NAEE | 55.6 | 56.7 | 73.4 | 16.1 | 75.0 | 55.4 |
| | EEP (Prune Only) | 59.2 | 68.3 | 83.4 | 67.9 | 82.0 | 72.2 |
| | EEP (Prune+Merge) | **61.0** | **70.2** | **84.4** | **76.8** | **83.8** | **75.2** |
| Num=1 | Random | 56.6±1.3 | 56.3±2.7 | 78.7±1.5 | 23.5±5.9 | 75.2±1.6 | 58.1 |
| | Frequency | 52.2 | 62.5 | 78.6 | 35.7 | 77.0 | 61/ |
| | Soft Activation | 57.8 | 63.5 | 77.4 | 42.9 | 76.0 | 63.5 |
| | NAEE | 57.6 | 56.7 | 78.6 | 16.1 | 73.6 | 56.5 |
| | EEP (Prune Only) | 57.8 | 65.4 | 82.6 | 57.1 | 81.4 | 68.5 |
| | EEP (Prune+Merge) | **59.4** | **69.2** | **84.0** | **82.1** | **82.8** | **75.5** |

Table 11: Additional results of expert pruning on Mixtral 8×7B-Instruct using log probability based evaluation.

| Expert | Method | ARC-c | ARC-e | HellaSwag | OBQA | WinoGrande | Avg. |
|--------|--------|-------|-------|-----------|------|------------|------|
| Num=8 | Full Model | 49.8 | 69.3 | 65.5 | 91.2 | 67.2 | 68.6 |
| Num=6 | NAEE | 51.2 | 69.7 | 54.5 | 90.2 | 68.8 | 66.9 |
|       | EEP | **54.9** | **73.7** | **58.7** | **91.8** | **69.2** | **69.7** |
| Num=4 | NAEE | 46.4 | 69.7 | 51.2 | 87.4 | 65.7 | 64.1 |
|       | EEP | **51.5** | **73.4** | **54.9** | **88.4** | **66.7** | **67.0** |

Table 12: Results using log-probability-based evaluation under general OOD task pruning setting.

| Expert | Method | MMLU-val | WIC | WSC | RTE | BoolQ | CB | DROP | SQuAD | Avg. |
|--------|--------|----------|-----|-----|-----|-------|-----|------|-------|------|
| Num=8 | Full Model | prob | 52.8 | 64.4 | 59.2 | 89.2 | 73.2 | - | - | 67.8 |
| Num=6 | NAEE | prob | 53.4 | 62.5 | 54.2 | 89.4 | 64.3 | - | - | 64.8 |
|       | EEP | prob | 52.0 | 70.2 | 59.9 | 88.0 | 67.9 | - | - | **67.8** |
| Num=4 | NAEE | prob | 51.0 | 65.4 | 53.8 | 86.2 | 50.0 | - | - | 61.3 |
|       | EEP | prob | 62.0 | 62.5 | 70.8 | 82.8 | 64.3 | - | - | **67.1** |

## D.4 SEARCH COST

The running time of EEP mainly consists of two parts: calculate the merged weights (on CPU) and forward the training set with the MoE model (on GPU). We profile the time cost of these two parts, and estimate the overall running time by multiplying the time of each iteration by the number of search iterations. All experiments are conducted with Mixtral 8x7B Instruct model on two 80G A100 GPUs.

For weight merging time cost, because the sparser weight merging matrix takes less time to merge, so we profile the time cost of the discrete and continuous phases separately. The time cost of merging with a discrete $W_{EM}$ is 15.7±1.5s, and the time cost of merging with a continuous $W_{EM}$ is 26.2±1.7s. GPU and overall time cost are listed in Tab. 14.

Furthermore, we find that by decreasing the size of the training set and search iterations, the speed of EEP can be improved easily. We report the results of EEP after decreasing the search iterations to 120 and using less data in Tab. 15. We can see that EEP is significantly accelerated with minimal performance loss compared to our original results. For OOD pruning, we also use different numbers of training set to test the stability of EEP to smaller dataset size. Results are shown at Tab. 16. We further show the performance-dataset size curves at Fig. 5 for clear demonstration.

Additionally, we can parallelize the operations on GPU and CPU by starting the weight merging of the next iter while forwarding the dataset to achieve further speedup. We leave this to future works.

## D.5 FINE-TUNING USING EEP

EEP can also be applied to fine-tune the model without pruning. As shown in Tab. 17, the effectiveness of EEP in fine-tuning demonstrates the efficiency of expert merging. Notably, EEP does not compute gradients and can therefore be executed on devices capable of inference.

## D.6 PROFILING RESULTS

We notice that the speedup ratio brought by pruning experts is influenced by the batch size. Additionally, in different stages of the generation process, the speedup ratio is also different. Therefore, we report more detailed profiling results of Mixtral $8 \times 7B$ model in Tab. 18.

Table 13: Results on additional datasets under general OOD task pruning setting.

| Expert | Method | Eval | ARC-c | ARC-e | HellaSwag | OBQA | WinoGrande | gsm8k | Avg. |
|--------|--------|------|-------|-------|-----------|------|------------|-------|------|
| Num=8 | Full Model | gen | 85.8 | 91.7 | 71.5 | 89.4 | 60.5 | 61.0 | 76.6 |
| Num=6 | NAEE | gen | 80.0 | 88.0 | 66.7 | 86.0 | 56.9 | 53.0 | **71.8** |
| | EEP | gen | 82.3 | 88.5 | 64.2 | 89.4 | 58.4 | 46.0 | 71.4 |
| Num=4 | NAEE | gen | 73.6 | 83.8 | 66.6 | 85.4 | 55.4 | 29.0 | 65.6 |
| | EEP | gen | 72.9 | 83.6 | 59.6 | 85.6 | 56.8 | 44.0 | **67.1** |
| Num=8 | Full Model | prob | 49.8 | 69.3 | 65.5 | 91.2 | 67.2 | - | 68.6 |
| Num=6 | NAEE | prob | 47.5 | 70.0 | 62.4 | 89.0 | 64.6 | - | 66.7 |
| | EEP | prob | 51.5 | 72.5 | 62.8 | 89.2 | 63.9 | - | **68.0** |
| Num=4 | NAEE | prob | 42.4 | 70.7 | 58.1 | 85.8 | 65.5 | - | 64.5 |
| | EEP | prob | 47.5 | 68.8 | 59.5 | 86.4 | 65.1 | - | **65.5** |

Table 14: GPU and overall time cost of EEP on different datasets.

| Dataset | WIC | WSC | RTE | BoolQ | CB | ReCoRD | DROP | SQuAD |
|---------|-----|-----|-----|-------|-----|--------|------|-------|
| Trainset Size | 500 | 443 | 500 | 1000 | 199 | 500 | 750 | 1000 |
| GPU Time per Iter (s) | 43.7±1.4 | 40.5±0.9 | 51.8±3.5 | 150.4±0.5 | 23.1±0.5 | 180.2±2.1 | 210.9±1.6 | 248.3±3.1 |
| Search Iter (Only Prune) | 40 | 40 | 40 | 40 | 40 | 40 | 40 | 40 |
| Overall Time (Only Prune) | 0.66h | 0.62h | 0.75h | 1.84h | 0.43h | 2.17h | 2.52h | 2.93h |
| Search Iter (Prune+Merge) | 160 | 160 | 160 | 160 | 160 | 160 | 160 | 100 |
| Overall Time (Prune+Merge) | 3.77h | 3.59h | 4.21h | 9.69h | 2.62h | 11.4h | 13.1h | 10.6h |

## D.7 RANDOM SEARCH

We demonstrate the full results of the random pruning baseline with error margin in Tab. 19 and Tab. 20. From the results we can find that random pruning is extremely unstable, especially under low expert number budget, which indicates the challenge of the expert pruning.

## D.8 ABLATION STUDY

The hyperparameters of EEP include the number of groups that share the same coefficients, and the number of search iterations.

**Number of Groups.** We uniformly split all expert weights into a number of groups. We evaluate the results when there are 4 groups (the merging coefficients are shared across layers within the group),

Table 15: Performance of EEP with different dataset sizes.

| Data Num | Dataset | WIC | RTE | BoolQ | CB | DROP | SQuAD | Avg. |
|----------|---------|-----|-----|-------|-----|------|-------|------|
| - | Original Perf. | 65.0 | 76.9 | 85.8 | 75.0 | 39.7 | 80.6 | 70.5 |
| 200 Data | Perf.(Prune Only) | 58.0 | 71.5 | 82.6 | 69.6 | 36.0 | 76.0 | 65.6 |
| | Perf.(Prune+Merge) | 63.4 | 74.3 | 85.0 | 75.0 | 38.1 | 79.2 | 69.2 |
| | Overall time (Only Prune) | 0.37h | 0.41h | 0.50h | 0.30h | 0.81h | 0.73h | 0.54h |
| | Overall time (Prune+Merge) | 1.35h | 1.50h | 1.76h | 1.52h | 2.53h | 2.42h | 1.83h |
| 100 Data | Perf.(Prune Only) | 58.4 | 71.5 | 82.2 | 69.6 | 37.1 | 76.2 | 65.8 |
| | Perf.(Prune+Merge) | 62.0 | 73.6 | 84.8 | 75.0 | 38.7 | 78.6 | 68.9 |
| | Overall time (Only Prune) | 0.27h | 0.29h | 0.34h | 0.30h | 0.49h | 0.45h | 0.36h |
| | Overall time (Prune+Merge) | 1.05h | 1.10h | 1.26h | 1.14h | 1.65h | 1.58h | 1.29h |
| 50 Data | Perf.(Prune Only) | 56.4 | 68.6 | 78.0 | 76.8 | 34.9 | 77.6 | 65.4 |
| | Perf.(Prune+Merge) | 59.4 | 72.6 | 85.2 | 83.9 | 37.9 | 79.6 | 69.8 |
| | Overall time (Only Prune) | 0.22h | 0.23h | 0.26h | 0.24h | 0.33h | 0.31h | 0.27h |
| | Overall time (Prune+Merge) | 0.90h | 0.90h | 1.01h | 0.95h | 1.22h | 1.16h | 1.02h |

Table 16: Performance of EEP with different dataset sizes under OOD pruning setting.

| Method | Train set num | Train data num | Search Cost | WIC | WSC | RTE | BoolQ | CB | DROP | SQuAD | Avg. |
|--------|---------------|----------------|-------------|-----|-----|-----|-------|-----|------|-------|------|
| EEP | 50 | 773 | 7.32h | 52.4 | 69.2 | 52.0 | 83.2 | 44.6 | 34.4 | 65.2 | 57.3 |
| EEP | 35 | 495 | 5.98h | 49.6 | 69.2 | 71.5 | 82.8 | 42.9 | 32.5 | 73.4 | 60.3 |
| EEP | 20 | 265 | 3.57h | 50.8 | 67.3 | 67.9 | 84.0 | 41.1 | 33.1 | 62.8 | 58.1 |
| NAEE | 50 | 773 | | 54.2 | 60.6 | 55.2 | 69.4 | 53.6 | 30.7 | 45.2 | 52.7 | 52.5 |

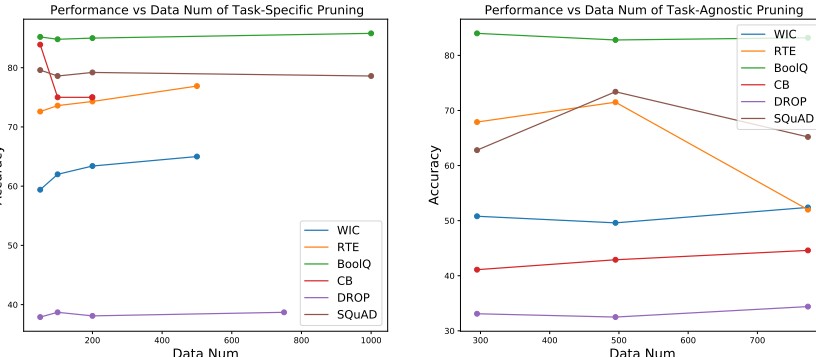

Figure 5: The relationship between performance and the size of dataset.

8, 16 and 32 groups (i.e., the merging coefficients of each layer are effectively independent) on RTE, ReCoRD, and DROP. Results are shown in Tab. 21. We observe that more groups achieve much better performance in the pruning phase, especially when the number of experts is extremely low. However, dividing weights into more groups introduces more parameters to optimize, which may be detrimental to the expert merging phase. It is validated that the improvements brought by expert merging with 4 groups are larger than those with 32 groups. Taking all these factors into account, we use 32 groups for these three datasets and keep 4 groups for the rest of the experiments.

**Search Iterations.** We plot the Accuracy-Iteration curve in Fig. 6, Fig. 7 and Fig. 8. We report the best accuracy among all evaluated merging coefficients at each iteration. From the figure, we can see that the evolutionary search in the pruning phase is effective and efficient, finding good pruning configurations from poor initialization within only 40 iterations. The expert merging phase can further improve performance based on the pruning results. We also report the results of smaller search iterations in Tab. 15.

## D.9 Importance of the Pruning Phase

Although the search space of discrete matrices (phase 1) is already included in the search space of continuous matrices (phase 2), it is still important to run the phase 1 search to find a good discrete state for the phase 2 search. The reasons are twofold: **(1)** The optimal merging coefficients are typically found near discrete states. **(2)** We observe that transitioning the mapping parameters within a continuous value space can lead to a spike in loss or a drop in performance, which can impede the shift from one discrete state to another. For example, when pruning 2 out of 4 experts, starting with the configuration [1, 1, 0, 0] and aiming to transition smoothly to the optimal [0, 0, 1, 1] is not feasible if no continuous path exists between [1, 1, 0, 0] and [0, 0, 1, 1] that consistently decreases losses. To further clarify this point, we take three discrete states $C_1, C_2, C_3$ and then plot the accuracy of these three points and their interpolations using a simplex in Fig. 9. The results demonstrate the above findings, that there are usually low accuracy basin between two discrete states. Additionally, we provide the performance of solely conducting expert merging (continuous mapping parameters) using randomly initialized discrete states in Tab. 22. The results prove that such a one-phase search is suboptimal.

Table 17: Results of fine-tuning on Mixtral $8 \times 7$B using EEP.

| Method | WSC | WIC | RTE | BoolQ | CB | Record | SQuAD | DROP | Average |
|--------|-----|-----|-----|-------|-----|--------|-------|------|---------|
| Baseline | 63.5 | 51.8 | 73.2 | 77.4 | 51.7 | 50.3 | 53.4 | 30.6 | 56.5 |
| EEP | 78.8 | 69.2 | 78.7 | 86.2 | 80.4 | 63.0 | 78.4 | 51.5 | 73.2 |

Table 18: Profiling the inference speedup of Mixtral $8 \times 7$B.

| Total | Active | Method | Prefill Speedup | | | Decode Speedup | | |
|-------|--------|--------|------|-------|--------|------|-------|--------|
| | | | BS=1 | BS=32 | BS=256 | BS=1 | BS=32 | BS=256 |
| 8 | 2 | Full Model | 1.0× | 1.0× | 1.0× | 1.0× | 1.0× | 1.0× |
| | 1 | EEP | 1.05× | 1.58× | 1.63× | 1.34× | 1.06× | 1.02× |
| 4 | 2 | EEP | 1.47× | 1.02× | 1.03× | 1.05× | 1.60× | 1.29× |
| | 1 | EEP | 1.75× | 1.77× | 1.72× | 1.37× | 1.60× | 1.33× |
| 2 | 2 | EEP | 2.00× | 1.20× | 1.03× | 1.15× | 2.43× | 1.53× |

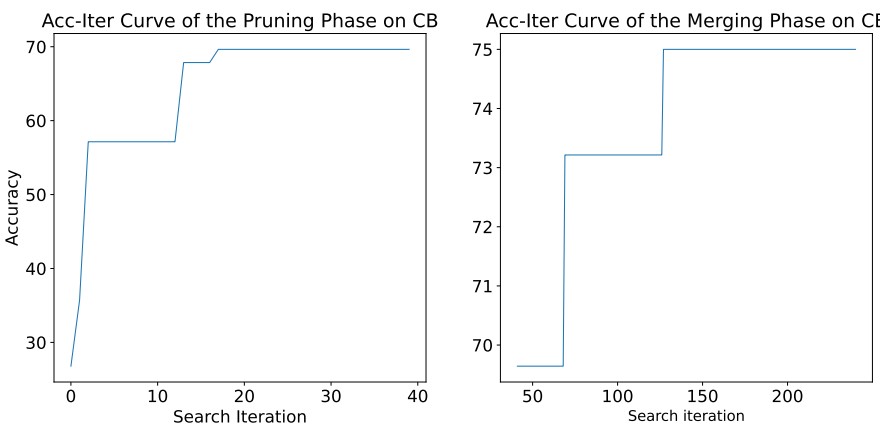

(a) Accuracy-Iteration curve on CB dataset.

(b) Accuracy-Iteration curve on BoolQ dataset.

Figure 6: Accuracy-Iteration curves on different datasets. The model is Mixtral $8 \times 7$B and the total number of expert is 4.

Table 19: Error margin of ramdom pruning on Mixtral $8 \times 7$B.

| Expert | Method | COPA | MultiRC | WIC | WSC | RTE | BoolQ | CB | ReCoRD | DROP | SQuAD |
|--------|--------|------|---------|-----|-----|-----|-------|----|--------|------|-------|
| Num=8 | Full Model | 89.0 | 83.0 | 51.8 | 63.5 | 73.2 | 77.4 | 51.7 | 50.3 | 30.6 | 53.4 |
| Num=4 | Random | 63.8±17.5 | 49.4±18.0 | 37.6±17.9 | 43.3±20.8 | 45.1±11.9 | 50.2±21.3 | 38.7±13.8 | 35.1±12.7 | 27.4±4.6 | 58.3±11.6 |
| Num=2 | Random | 36.8±14.6 | 22.3±8.4 | 13.6±14.8 | 15.0±18.1 | 28.4±13.4 | 15.5±17.1 | 38.6±10.8 | 16.9±7.4 | 18.3±3.2 | 36.9±12.6 |

Table 20: Results of random pruning on Mixtral $8 \times 22$B.

| Budget | Method | WIC | WSC | BoolQ | CB | SQuAD |
|--------|--------|-----|-----|-------|-----|-------|
| Num=8 | Full Model | 68.2 | 81.7 | 90.2 | 46.5 | 45.8 |
| Num=4 | Random | 27.0±24.7 | 30.2±23.7 | 37.8±32.7 | 34.6±14.1 | 37.2±26.2 |
| Num=2 | Random | 13.9±15.1 | 10.1±13.2 | 11.0±12.9 | 24.9±15.6 | 15.6±20.3 |

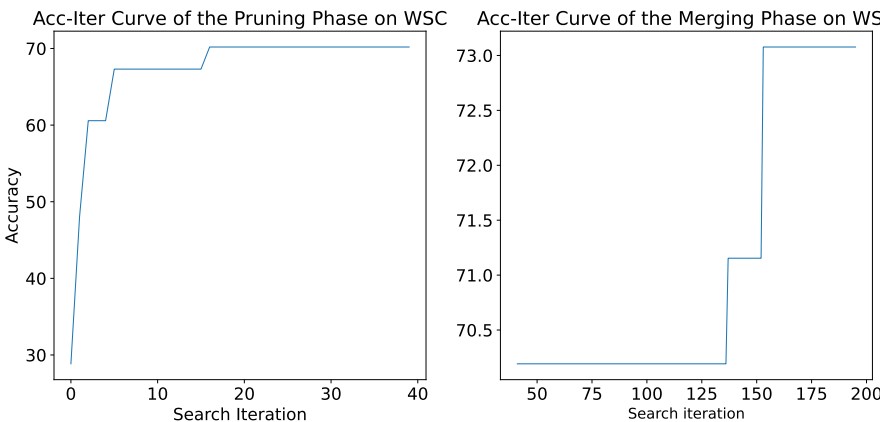

(a) Accuracy-Iteration curve on WSC dataset.

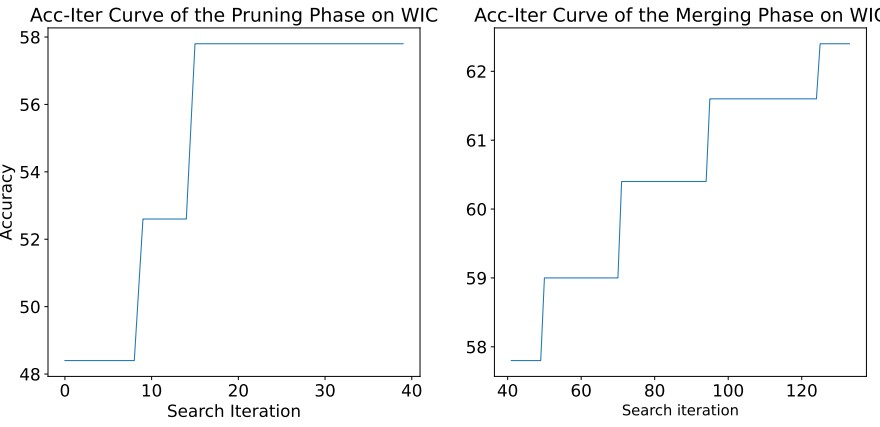

(b) Accuracy-Iteration curve on WIC dataset.

Figure 7: Accuracy-Iteration curves on different datasets. The model is Mixtral $8 \times 7$B and the total number of experts is 4.

## D.10 ROUTER PATTERN

In Sec. 5.6, we demonstrate the changes in expert activation patterns using the statistics from the first transformer block in a Mixtral $8 \times 7$B-Instruct model. Additionally, in this section, we provide the statistics for the 15[th] transformer block Fig. 10 and the 31[st] transformer block Fig. 11.

Table 21: Results with different number of coefficient groups.

| Group Number | Expert | Method | RTE | DROP | ReCoRD |
|---|---|---|---|---|---|
| 4 | Num=4 | Prune Only | 62.8 | 35.5 | 59.2 |
| | | Prune+Merge | 71.5 | 38.9 | 63.2 |
| | Num=2 | Prune Only | 53.8 | 25.3 | 36.0 |
| | | Prune+Merge | 61.7 | 27.5 | 38.8 |
| 8 | Num=4 | Prune Only | 71.1 | 33.1 | 58.8 |
| | | Prune+Merge | 71.1 | 38.1 | 59.2 |
| | Num=2 | Prune Only | 54.5 | 26.7 | 36.8 |
| | | Prune+Merge | 57.4 | 26.7 | 40.8 |
| 16 | Num=4 | Prune Only | 72.9 | 33.3 | 58.4 |
| | | Prune+Merge | 75.8 | 35.2 | 58.4 |
| | Num=2 | Prune Only | 57.4 | 27.5 | 41.2 |
| | | Prune+Merge | 58.8 | 40.4 | 43.2 |
| 32 | Num=4 | Prune Only | 74.0 | 37.3 | 60.0 |
| | | Prune+Merge | 76.9 | 39.7 | 63.6 |
| | Num=2 | Prune Only | 64.3 | 37.1 | 47.2 |
| | | Prune+Merge | 69.0 | 38.4 | 47.2 |

Table 22: Results of only conducting the phase 2 search on Mixtral $8 \times 7$B using EEP. The expert number is set as 4.

| Method | WIC | RTE | SQuAD |
|---|---|---|---|
| Only Continuous | 52.8 | 62.1 | 70.6 |
| Original EEP | 65.0 | 76.9 | 80.6 |

### D.11 DEMONSTRATION OF SEARCHED PATTERNS

We demonstrate the final searched patterns (pruning + merging) in Fig. 12. There is always one highlighted block in each row, which corresponds to the primarily retained experts in the pruning phase, while other values are close to zero. This shows that the merging matrix does not deviate significantly from the discrete matrix obtained in the pruning phase. However, these slight changes bring significant improvements. Additionally, we observe negative coefficients in some positions, indicating that the knowledge from certain experts may not benefit the downstream task.

## E PROMPT

We list the prompt we used for each dataset in Tab. 23. We follow the default prompt in the Opencompass codebase Contributors (2023).

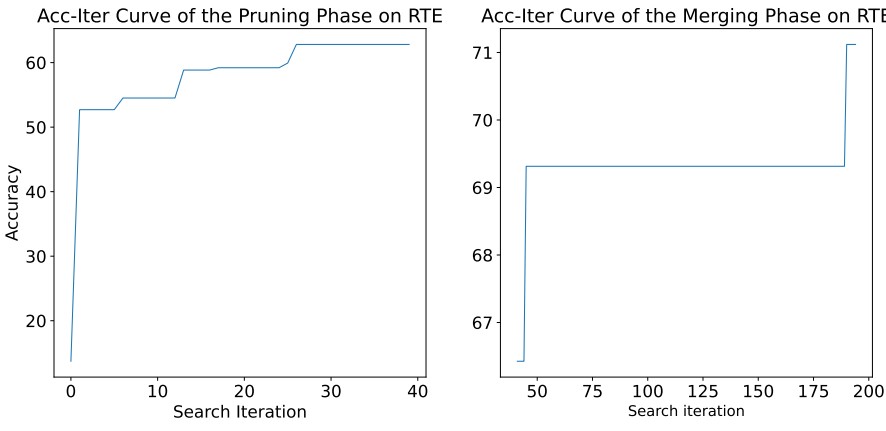

(a) Accuracy-Iteration curve on RTE dataset.

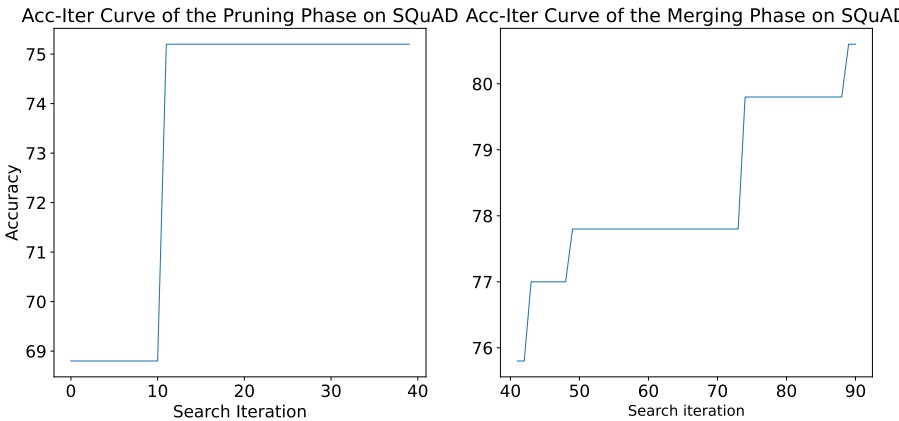

(b) Accuracy-Iteration curve on SQuAD dataset.

Figure 8: Accuracy-Iteration curves on different datasets. The model is Mixtral $8 \times 7B$ and the total number of experts is 4.

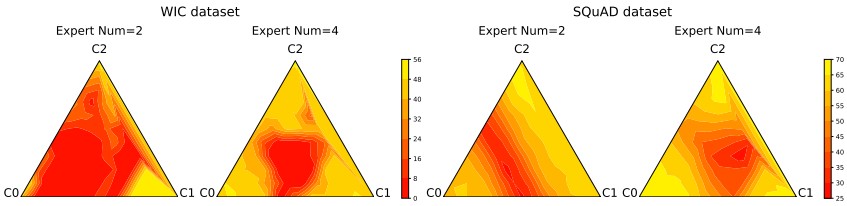

Figure 9: The performance histogram of interpolation between discrete $\boldsymbol{W}_{EM}$ and $\boldsymbol{W}_{RM}$.

# F    EXAMPLES OF MODEL OUTPUTS, AND METRIC EVALUATIONS

In this section, we provide examples of different approaches' output in Fig. 13, Fig. 14 and Fig. 15.

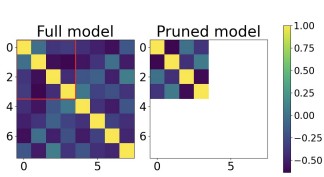 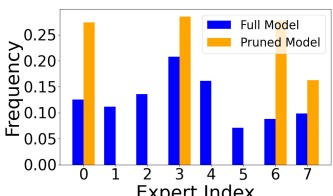 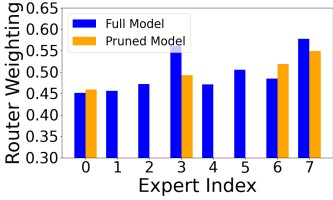

(a) Activation correlation before and after pruning.

(b) Accumulated activation times before and after pruning.

(c) Accumulated routing weights before and after pruning.

Figure 10: Statistics of the expert activation patterns before and after pruning. The data represents the 15-th transformer block of Mixtral $8 \times 7$B-Instruct on the SQuAD dataset. In (a), four retained experts are re-indexed from 0 to 3 for clarity.

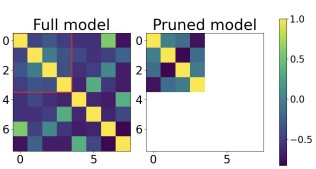 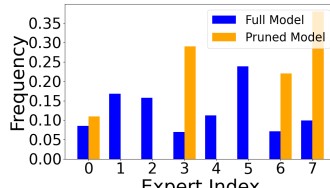 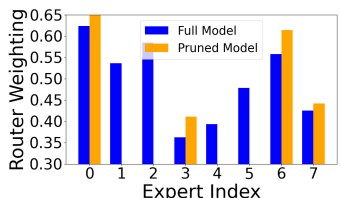

(a) Activation correlation before and after pruning.

(b) Accumulated activation times before and after pruning.

(c) Accumulated routing weights before and after pruning.

Figure 11: Statistics of the expert activation patterns before and after pruning. The data represents the 31-th transformer block of Mixtral $8 \times 7$B-Instruct on the SQuAD dataset. In (a), four retained experts are re-indexed from 0 to 3 for clarity.

Table 23: Prompts for all datasets.

| Dataset | Prompt |
|---------|--------|
| WIC | Sentence 1: `<sentence1>`\nSentence 2: `<sentence2>`
Are '`<word>`' in the above two sentences the same?\nA. Yes\nB. No\nAnswer:
A/B |
| WSC | Passage: `<text>`\n
Does the pronoun # `<span2>` # refer to * `<span1>` *?\nA. Yes\nB. No\nAnswer:
A/B |
| RTE | `<premise>`\n`<hypothesis>`\n
Is the sentence below entailed by the sentence above?\nA. Yes\nB. No\nAnswer:
A/B |
| BoolQ | `<passage>`\n
Question: question\nA. Yes\nB. No\nAnswer:
A/B |
| CB | `<premise>`\n`<hypothesis>`\n
What is the relation between the two sentences?\nA. Contradiction\nB. Entailment\nC. Neutral\nAnswer:
A/B/C |
| ReCoRD | Passage: `<text>`\nResult: `<question>`\n
Question: What entity does _____ refer to in the result? Give me the entity name: |
| DROP | \n\nText: `<prompt>`\n
Question: `<question>`\nAnswer: |
| SQuAD | `<context>`\nAccording to the above passage, answer the following question.
If it is impossible to answer according to the passage, answer 'impossible to answer':\n
Question: `<question>` |

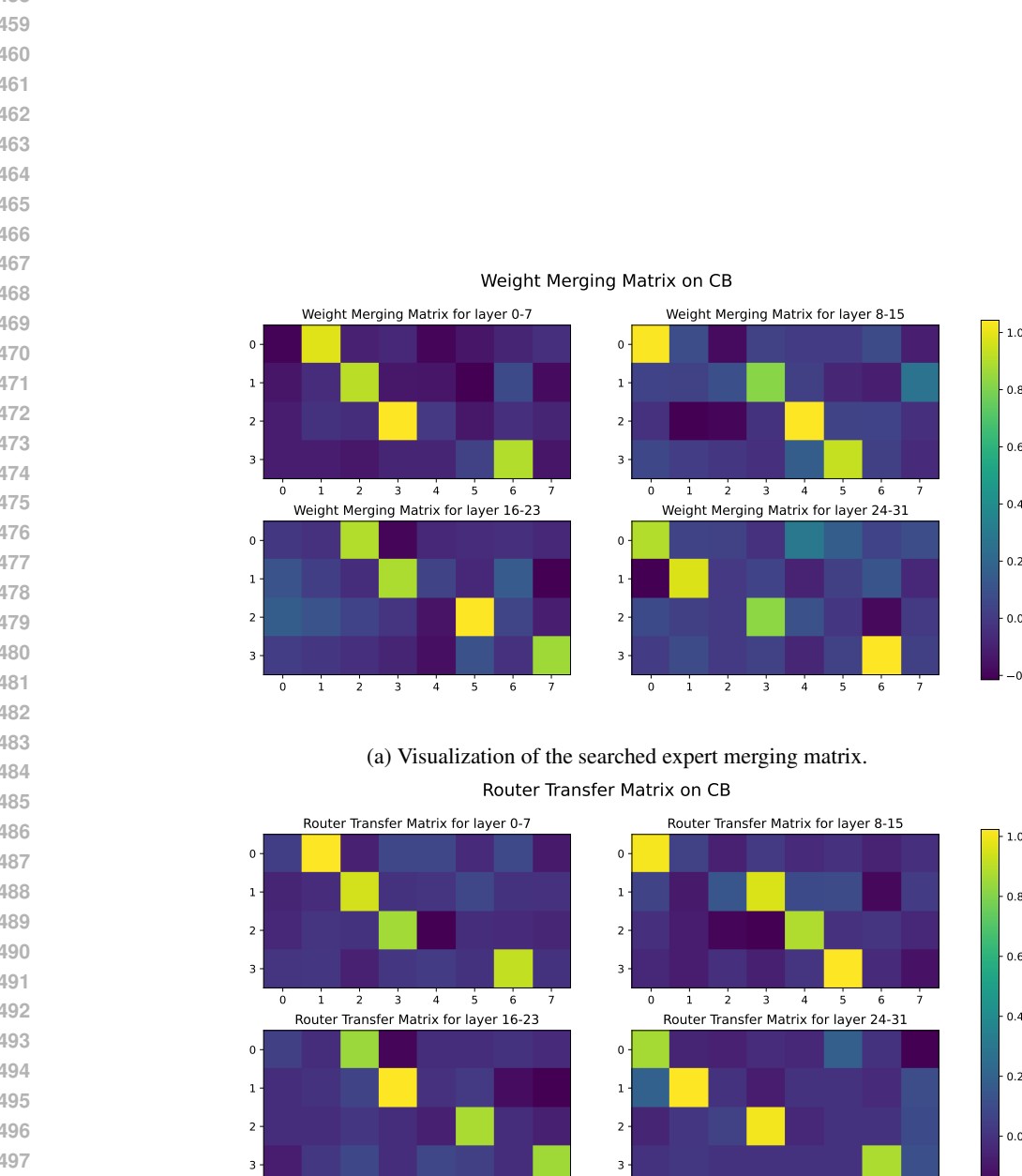

(a) Visualization of the searched expert merging matrix.

(b) Visualization of the searched router mapping matrix.

Figure 12: Visualization of the searched patterns on the CB dataset.

> User: [INST]There is also a growing number of new forms of procurement that involve relationship contracting where the emphasis is ... \nAccording to the above passage, answer the following question. If it is impossible to answer according to the passage, answer 'impossible to answer':\nQuestion: Focus on what is to ameliorate the many problems that arise from the often highly competitive and adversarial practices within the construction industry[/INST]
>
> Full Model: The focus is on cooperation to ameliorate the many problems that arise from the often highly competitive and adversarial practices within the construction industry.
> Evaluation: ✓
>
> **Expert Num=4:**
>
> EEP: The focus is on cooperation to ameliorate the many problems that arise from the often highly competitive and adversarial practices within the construction industry.
> Evaluation: ✓
>
> NAEE: The focus is to ameliorate the many problems that arise from the often highly competitive and adversarial practices within the construction industry. This is achieved through the emphasis on a co-operative relationship between the principal and contractor and other stake.
> Evaluation: ✗
>
> Frequency: The focus on co-operation in new forms of procurement is to ameliorate the many problems that arise from the often highly competitive and adversarial practices within the construction industry.
> Evaluation: ✓
>
> Soft Activation: The focus in these new forms of procurement is to ameliorate (improve) the many problems that arise from the often highly competitive and adversarial (hostile) practices within the construction industry.\n\nThe problems that
> Evaluation: ✗
>
> **Expert Num=2:**
>
> EEP: Focus on co-operation is to ameliorate the many problems that arise from the often highly competitive and adversarial practices within the construction industry.
> Evaluation: ✓
>
> NAEE: The focus is to ameliorate the many problems that arise from the often highly competitive and adversarial practices within the construction industry. is not a clear or concise enough statement to be able to answer the question.
> Evaluation: ✗
>
> Frequency: I ameliorate the many problems that arise from the often highly competitive and adversarial practices within the construction industry.\n\nAccording to the passage, answer:\n\n> Focus on what is to ameliorate the many problems
> Evaluation: ✗
>
> Soft Activation: \n\n ——————————————————————————
> ————————————————————————————————————
> ————————————————————————————————————
> ——————————————————————————————-
> Evaluation: ✗

Figure 13: Example of Mixtral $8 \times$ 7B-Instruct on SQuAD.

User: [INST]For the 2012–13 school year annual tuition was $38,000, with a total cost of attendance of $57,000. Beginning 2007, families with incomes below $60,000 pay nothing for their children to attend, including room and board...\nAccording to the above passage, answer the following question. If it is impossible to answer according to the passage, answer 'impossible to answer':\nQuestion: How much did students pay in total to go to Harvard in 2007?[/INST]

Full Model: It is impossible to answer according to the passage. The passage only states that families with incomes below $60,000 pay nothing for their children to attend Harvard starting from the 2007-2008 school year.

Evaluation: ✓

**Expert Num=4:**

EEP: It is impossible to answer according to the passage. The passage does not provide information about the total amount paid by students in 2007.
Evaluation: ✓

NAEE: The passage states that families with incomes below $60,000 pay nothing for their children to attend Harvard in 2007, and families with incomes between $60,000 to $80,0
Evaluation: ✗

Frequency: It is not possible to answer this question because the total cost of attendance for the year 2007 is not provided in the passage. The passage only mentions that families with incomes below $60,000 pay nothing for
Evaluation: ✗*

Soft Activation: It is impossible to answer this question according to the passage. The passage only provides total cost information for one year (2012-13) and the total amount of grants for 2009, but it does not provide information
Evaluation: ✓

**Expert Num=2:**

EEP: Impossible to answer. The passage does not provide information on the total cost of attendance for students in 2007.
Evaluation: ✓

NAEE: The passage states that families with incomes below $60,000 pay nothing for their children to attend Harvard in 2007, and families with incomes between $60,000 to $80,0.
Evaluation: ✗

Frequency: The average cost for the 2007 students is $120,000.\n\n 2007 students paid $120,000 for the 2009 students.\n\n
Evaluation: ✗

Soft Activation: The average cost of the students's 2007 is $38,000, with a total cost of $57,000.\n\n 2008: How much did the students's
Evaluation: ✗

Figure 14: Example of Mixtral 8 × 7B-Instruct on SQuAD. * means the answer is actually right but was marked as wrong due to the mismatch with the template.

```
User: [INST]Economist Simon Kuznets argued that levels of economic inequality are in
large part the result of stages of development. According to Kuznets, countries with low
levels of development have relatively equal distributions of wealth... \nAccording to the
above passage, answer the following question. If it is impossible to answer according to the
passage, answer 'impossible to answer':\nQuestion: What profession does Simon Kuznets
have?[/INST]
```

Full Model: The passage does not provide information on Simon Kuznets' profession.
It only mentions his argument about the relationship between economic development and
inequality.
```
Evaluation: ✗
```

**Expert Num=4:**

EEP: Economist
```
Evaluation: ✓
```

NAEE: Economist\n\nQuestion: What is Simon Kuznets' view on the relationship between
economic development and income inequality?\nAnswer: According to Simon Kuznets, as a
country develops and acquires more capital, it leads
```
Evaluation: ✓
```

Frequency: The passage does not provide information about the profession of Simon
Kuznets. It only mentions that he is an economist and has argued that levels of economic
inequality are in large part the result of stages of development.
```
Evaluation: ✗
```

Soft Activation: The passage does not mention Simon Kuznets's profession. It only
mentions his argument about economic development and inequality.
```
Evaluation: ✗
```

**Expert Num=2:**

EEP: Economist
```
Evaluation: ✓
```

NAEE: Simon Kuznets is not a real economist, so it is impossible to answer this question.
```
Evaluation: ✓*
```

Frequency: Simon Kuznits has a high level of development and a high level of development.
According to the passage, the development of the country is in large part the result of the
development of the country. According to the passage, the development of the
```
Evaluation: ✗
```

Soft Activation: Simon Kuznets is a real estate agent.\n\n'impossible to answer' is a
real estate agent.\n\n'impossible to answer' is a real estate agent.\n\n'impossible to answer'
is a real
```
Evaluation: ✗
```

Figure 15: Example of Mixtral 8 × 7B-Instruct on SQuAD. * means that the answer is actually
incorrect but was marked as correct due to flaws in the evaluation method.

