# OpenReview forum: "Efficient Expert Pruning for Sparse Mixture-of-Experts Language Models: Enhancing Performance and Reducing Inference Costs"
_ICLR.cc/2025/Conference — Submitted to ICLR 2025_

### Official Review · Reviewer_N5TL · 2024-10-30

**Soundness:** 3
**Presentation:** 3
**Contribution:** 3
**Rating:** 6
**Confidence:** 3

**Summary:**

This paper proposes Efficient Expert Pruning (EEP), a gradient-free evolutionary strategy to optimize the pruning and merging of experts within Sparse Mixture-of-Experts (SMoE) models. EEP aims to enhance performance while saving GPU memory and reducing inference costs by minimizing both the total number of experts and the number of active experts during inference. This is achieved through an evolutionary search process that optimizes two matrices: a router mapping matrix for expert selection and an expert merging matrix to retain pruned experts' knowledge. EEP is evaluated across various downstream tasks and shows considerable improvement over baselines in terms of memory efficiency, inference speed, and overall performance.

**Strengths:**

1. This paper is well-written, well-organized including problem formulation, methodological exposition, and experimental setup, making it accessible to readers with a background in the field. Besides, the implementation details and code are provided for reproducing.
2. The EEP method employs a gradient-free evolutionary search strategy for expert pruning and merging in the SMoE model, which is a relatively novel method.
3. This paper conducts extensive experiments on multiple datasets and models to validate the effectiveness of the method, demonstrating significant improvements over existing baselines, which greatly enhances the practicality of SMoE pruning.

**Weaknesses:**

1. In the NAEE [1] method, the authors use the C4 dataset as the calibration dataset and evaluate the zero-shot accuracy of the pruned model, which is a common experimental setup in the field of LLM compression. The authors should conduct similar experiments and provide data in this regard to allow for a more comprehensive assessment of the effectiveness of the EEP method.
2. I am concerned about the search cost of EEP. The total time spent on EEP for each dataset is several hours, and this is just for the smallest Mixtral 8x7B Instruct model. For larger datasets and larger models, EEP will take even longer, which is often unacceptable.
3. The out-of-distribution accuracy of EEP is much lower than that of NAEE [1] on many datasets (Tables 5, 12 and 13), so I am concerned about the generalization ability of EEP.
4. Some notations are confusing.
    - The authors use bold uppercase symbols to represent matrices. In line 91, $W_{RM}$ and $W_{EM}$ should be bolded to ensure consistency with the notation used in Section 4.2.
    - The symbol $\boldsymbol{W}_G$ in Equation 2 is not explained. I suspect it refers to the router network $\boldsymbol{W}_R$ in line 179.

[1] Not all experts are equal: Efficient expert pruning and skipping for mixture-of-experts large language models. ACL 2024 Main.

**Questions:**

1. In Section 5.1, the authors state that the dataset is randomly divided into training and test set, with the training set used for evolutionary search and the test set used for evaluation. How exactly is this division performed? Table 14 seems to contain some data on the trainset size, but it is not comprehensive enough.
2. The authors used 50 MMLU datasets as the training set and 7 as the test set, utilizing a large number of datasets as the training set to test accuracy on out-of-distribution datasets. I would like to know the total search time EEP took in this case, and whether using fewer training sets would impact the out-of-distribution accuracy.
3. How does the accuracy of the EEP pruned model obtained from one dataset perform on other datasets? For example, if BoolQ is used as the training set, what is the accuracy of the pruned model on other validation sets?

---

> ### Author Response · Authors · 2024-11-27
> **Response to Reviewer N5TL 1**
>
> We greatly appreciate the reviewer’s constructive feedback and insightful suggestions, which have been instrumental in helping us enhance our paper. Below, we address your questions one-by-one.
>
> **W1:** In the NAEE [1] method, the authors use the C4 dataset as the calibration dataset and evaluate the zero-shot accuracy of the pruned model, which is a common experimental setup in the field of LLM compression. The authors should conduct similar experiments and provide data in this regard to allow for a more comprehensive assessment of the effectiveness of the EEP method.}
>
> -  For our OOD pruning results, we adopt similar settings by first conducting pruning on a large dataset MMLU and then evaluating the results on downstream tasks. We did not use C4 as the calibration dataset because it is a dataset for pre-training where no metrics like accuracy can be counted on this dataset. As an alternative, we choose MMLU since it is a large and diverse dataset to prompt generalization.
>
> **W2:** I am concerned about the search cost of EEP. The total time spent on EEP for each dataset is several hours, and this is just for the smallest Mixtral 8x7B Instruct model. For larger datasets and larger models, EEP will take even longer, which is often unacceptable.
>
> - Though EEP is not designed for quick pruning, we would like to point out that the search cost of EEP can be easily reduced by using a smaller size of training sets---we have shown such results in Table  15 of our initial submission. To better illustrate the point,
> we provide more results below, including the performance and the search cost of EEP under 200, 100, and 50 data points. We can see that EEP (Prune only) with 50 data points can complete the pruning process in a very short time (**less than 20 minutes**) while having good performance.  Such computational cost is comparable with the baseline method NAEE, but our EEP shows significant performance improvement compared to both the full model and the baseline.
>
>     **Performance of EEP with Different Data Sizes**
>     |                        | WIC  | RTE  | BoolQ | CB   | DROP | SQuAD | Avg   |
>     |------------------------|------|------|-------|------|------|-------|-------|
>     | Original (Prune only)  | 57.8 | 74.0 | 82.8  | 69.6 | 37.3 | 75.2  | 66.1  |
>     | Original (Prune+Merge) | 65.0 | 76.9 | 85.8  | 75.0 | 38.7 | 78.6  | 70.5  |
>     | 200 data (Prune only)   | 58.0 | 71.5 | 82.6  | 69.6 | 36.0 | 76.0  | 65.6- |
>     | 200 data (Prune+Merge)  | 63.4 | 74.3 | 85.0  | 75.0 | 38.1 | 79.2  | 69.2  |
>     | 100 data (Prune only)   | 58.4 | 71.5 | 82.2  | 69.6 | 37.1 | 76.2  | 65.8  |
>     | 100 data (Prune+Merge) | 62.0 | 73.6 | 84.8  | 75.0 | 38.7 | 78.6  | 68.9  |
>     | 50 data (Prune only)    | 56.4 | 68.6 | 78.0  | 76.8 | 34.9 | 77.6  | 65.4  |
>     | 50 data (Prune+Merge)  | 59.4 | 72.6 | 85.2  | 83.9 | 37.9 | 79.6  | 69.8  |
>
>     **Running Time of EEP and NAEE**
>     |                 | WIC   | RTE   | BoolQ | CB    | DROP  | SQuAD |
>     |-----------------|-------|-------|-------|-------|-------|-------|
>     | NAEE            | 0.11h | 0.13h | 0.19h | 0.15h | 0.26h | 0.21h |
>     | EEP(Prune only) | 0.22h | 0.23h | 0.26h | 0.24h | 0.33h | 0.27h |
>
> - Additionally, in practice, the computational costs during deployment are often much higher than those during training. For example, OpenAI spent more than 4 billion dollars this year to run the inference workload of ChatGPT, while \$ 3 billion was spent to complete the full training of ChatGPT as well as other models (source: https://www.datacenterdynamics.com/en/news/openai-training-and-inference-costs-could-reach-7bn-for-2024-ai-startup-set-to-lose-5bn-report/).
> Therefore, we focused our evaluation on demonstrating how EEP improves the model efficiency (including GPU memory consumption and inference latency) during deployment, rather than its one-time search cost.
>
> -  We also want to emphasize that one main strength in the computational efficiency of EEP as a gradient-free method is that EEP can be conducted on devices that are only available for inference. In contrast, gradient-based approaches require substantially more GPU memory, which may be an important problem if the user has a limited GPU resources.

---

> ### Author Response · Authors · 2024-11-27
> **Response to Reviewer N5TL 2**
>
> **W3:** The out-of-distribution accuracy of EEP is much lower than that of NAEE [1] on many datasets (Tables 5, 12 and 13), so I am concerned about the generalization ability of EEP.
>
> - We would like to point out that among the 46 comparisons across two types of budget (6 and 4), two evaluation methods (generation-based and log-probability based), and 13 datasets, we failed to outperform NAEE in only 16 cases, achieving a win rate of 65\%. Moreover, our average accuracy also surpasses NAEE. Therefore, we believe that under the OOD pruning setting, our method is competitive to the baseline approach.
>
> - Regarding the phenomenon that EEP can not outperform the baseline on some datasets, we think it may be because the optimization objective of EEP is the accuracy on the training set, while the optimization objective of NAEE is the gap to the full model. As a result, EEP might have more reliance on the diversity of the training dataset. Even though MMLU is a diverse dataset that contains many different sub-tasks, it is natural that MMLU may not be fully aligned with some downstream tasks and this fact leads to worse performance on some of the datasets.
>
> -  Finally, We want to emphasize that EEP can achieve **more impressive results** on the task-specific compression scenario. We would like to point out the importance of this scenario:
>
>     - Task-specific pruning also has a lot of application scenarios. Task-specific LLMs fine-tuned from general capability LLMs, such as Code LLama, BioMistral, and Llama Guard, are widely used in many applications. Recent papers [1] also argue that composing different task-specific models/modules rather than using one general-purpose model and expecting it to perform well on all tasks might be better in terms of scalability, efficiency, and so on. Using EEP to adapt a general-purpose MoE model towards one specific downstream task while making it more efficient aligns with these trends.
>
> [1] Xiao et al. "Configurable foundation models: Building llms from a modular perspective."
>
> **W4:** Some notations are confusing.
> - We thank the reviewer for carefully reading our work. We have updated the notations in the revision.
>
> **Q1:** In Section 5.1, the authors state that the dataset is randomly divided into training and test set, with the training set used for evolutionary search and the test set used for evaluation. How exactly is this division performed? Table 14 seems to contain some data on the training set size, but it is not comprehensive enough.}
>
> - **How to conduct the division:** For each dataset, if the downloaded official version already has training and test sets separated, we will randomly draw a portion from each of them as our training and test sets. If the downloaded version only contains one large dataset, we will randomly sample two subsets from it as the training and test sets.
>
> - **More comprehensive information about dataset size:** We provide a more complete table for the information of each dataset, shown below.
>
>   **The number of samples in each dataset in Table 1**
>   |       | COPA | MultiRC | WIC | WSC | RTE | BoolQ | CB  | ReCoRd | DROP | SQuAD |
>   |-------|------|---------|-----|-----|-----|-------|-----|--------|------|-------|
>   | Train | 300  | 750     | 500 | 443 | 500 | 1000  | 199 | 500    | 750  | 1000  |
>   | Test  | 100  | 500     | 500 | 104 | 277 | 500   | 56  | 500    | 750  | 500   |
>
>   **The number of samples in each dataset in Table 2**
>   |       | Arc-c | Arc-e | HellaSwag | OBQA | WinoGrande | gsm8k |
>   |-------|-------|-------|-----------|------|------------|-------|
>   | Train | 300   | 300   | 300       | 300  | 300        | 400   |
>   | Test  | 300   | 600   | 600       | 600  | 600        | 200   |

---

> ### Author Response · Authors · 2024-11-27
> **Response to Reviewer N5TL 3**
>
> **Q2:** The authors used 50 MMLU datasets as the training set and 7 as the test set, utilizing a large number of datasets as the training set to test accuracy on out-of-distribution datasets. I would like to know the total search time EEP took in this case, and whether using fewer training sets would impact the out-of-distribution accuracy.
>
> - Following your suggestion, we conduct two additional experiments by using only 20 and 35 subsets randomly chosen from the original 50 subsets. Model performance and search time cost are shown in the tables below. Under the OOD setting, we can also find that EEP is very robust to the size of the dataset. Therefore, the search cost of EEP can be decreased easily.
>
>   |Method| Train set num | Train data num | Search Cost | WIC   | WSC   | RTE   | BoolQ | CB    | DROP  | SQuAD | Avg. |
>   |----------|---------------|----------------|-------------|-------|-------|-------|-------|-------|-------|-------|------|
>   |EEP| 50            | 773            | 7.32h       | 52.4  | 69.2  | 52.0  | 83.2  | 44.6  | 34.4  | 65.2  | 57.3 |
>   |EEP| 35            | 495            | 5.98h       | 49.6  | 69.2  | 71.5  | 82.8  | 42.9  | 32.5  | 73.4  | 60.3 |
>   |EEP| 20            | 265            | 3.57h       | 50.8  | 67.3  | 67.9  | 84.0  | 41.1  | 33.1  | 62.8  | 58.1 |
>   | NAEE     |50     | 773             | 1.24h        | 54.2  | 60.6  | 55.2  | 69.4  | 53.6  | 30.7  | 45.2  | 52.7 |
>
> - We have added these results to Table 16 and Figure 5 of our revision.
>
> - We also want to point out that, for the out-of-distribution pruning setting, a particularly large dataset is often used as the calibration dataset. Therefore, it is reasonable to use the whole MMLU, a large, diverse, and public dataset, to conduct EEP, as we did in the original submission.
>
> **Q3:** How does the accuracy of the EEP pruned model obtained from one dataset perform on other datasets? For example, if BoolQ is used as the training set, what is the accuracy of the pruned model on other validation sets?
>
> - Following your suggestion, we conduct the experiment and the results are provided below. Each column means different datasets for pruning, and each row means different datasets for evaluation.
>
>     |       | WSC  | WIC  | RTE  | BoolQ | DROP | SQuAD |
>     |-------|------|------|------|-------|------|-------|
>     | WSC   | 74.0 | 52.8 | 63.5 | 64.4  | 36.5 | 54.8  |
>     | WIC   | 49.2 | 64.2 | 47.8 | 50.2  | 48.8 | 5.8   |
>     | RTE   | 49.5 | 58.8 | 75.5 | 69.7  | 47.6 | 46.9  |
>     | BoolQ | 80.4 | 60.4 | 77.8 | 84.6  | 51.0 | 56.8  |
>     | DROP  | 30.4 | 31.7 | 28.0 | 30.1  | 37.1 | 24.8  |
>     | SQuAD | 70.2 | 67.0 | 64.0 | 64.4  | 67.0 | 76.2  |
>
> - We can see that the searched results from WSC, RTE, and BoolQ generalize better with each other, whereas the searched results from DROP and SQuAD have relatively worse performance on other datasets. This might be due to the similarity between different datasets: DROP and SQuAD are about generative tasks, whereas other datasets are about multi-choice selection.
>
> - These results suggest that conducting search only on a single dataset is not beneficial for generalization.
> We suggest users of EEP use large and diverse datasets to conduct EEP search, where we see much better generalization performance
> in Tables 5, 12, 13.

---

> ### Author Response · Authors · 2024-12-01
> **Follow-up Comment**
>
> Thank you for your insightful review. We sincerely appreciate your recognition of the clarity, novelty and effectiveness of EEP.
>
> Since your feedback has been crucial in refining our work, we have worked diligently in our rebuttal to address the concerns and questions raised. We hope our responses and additional clarifications have further strengthened your confidence in the contributions of our work.
>
> If you find that our response has satisfactorily resolved your concerns, we would kindly ask you to consider revisiting your score to better reflect your updated evaluation of the paper. We respect your discretion and remain grateful for your valuable feedback and initial positive evaluation.
>
> Thank you again for your time and insightful review. If you have any further questions, we are very happy to discuss.

---

> > ### Comment · Reviewer_N5TL · 2024-12-03
> >
> > Thank you for your responses. I will retain my score.

---

### Official Review · Reviewer_oSj6 · 2024-11-01

**Soundness:** 2
**Presentation:** 3
**Contribution:** 3
**Rating:** 3
**Confidence:** 2

**Summary:**

This paper addresses the challenge of pruning in sparse mixture of experts (SMoE) models. The authors propose a gradient-free evolutionary strategy called Efficient Expert Pruning (EEP) to improve the pruning process for experts within these models. Experimental results on diverse models and datasets demonstrate the effectiveness of this approach.

**Strengths:**

(1) The writing is clear and easy to follow.

(2) The proposed method is interesting, suggesting that evolutionary strategies can effectively optimize parameter combinations for model pruning.

(3) The insights presented are valuable, demonstrating that MoE models contain redundancy and hold significant potential for pruning while maintaining performance and improving efficiency.

**Weaknesses:**

(1) Lack of Reliable Evaluations: I find it puzzling that Mixtral-8x22B's average performance improves despite being pruned from 8 experts down to 2. This raises questions about the robustness of the evaluation methods, especially given the relatively small datasets used in this study. In contrast, NAEE—a key baseline—evaluates on the comprehensive MMLU dataset, which could serve as a more reliable benchmark.

(2) In Table 6, even after pruning from 8 experts down to 4 and then 2, the speedup is small. What factors might explain this limited increase in speed?

(3) The method’s efficiency is largely limited by its computational demands and the dataset used. The pruning dataset is relatively small, yet the process still requires significant time—up to several hours.

(4) Are the training and evaluation samples drawn from the same distribution? This could make sense as a task-agnostic pruning approach. However, given the long pruning time even with limited samples, why not fine-tune a roughly pruned model instead?

**Questions:**

see weakness

---

> ### Author Response · Authors · 2024-11-27
> **Response to Reviewer oSj6 1**
>
> We greatly appreciate the reviewer’s constructive feedback and insightful suggestions, which have been instrumental in helping us enhance our paper. Below, we address your questions one-by-one.
>
> **W1:** I find it puzzling that Mixtral-8x22B's average performance improves despite being pruned from 8 experts down to 2. This raises questions about the robustness of the evaluation methods, especially given the relatively small datasets used in this study. In contrast, NAEE—a key baseline—evaluates on the comprehensive MMLU dataset, which could serve as a more reliable benchmark.
>
>   - **Regarding the evaluation framework:** In this work we adopt generation-based evaluation following the implementation of OpenCompass (source: https://github.com/open-compass/opencompass) as our evaluation framework, which has been widely used. We adopt the default settings in OpenCompass to ensure less intervention in the evaluation.
>
>   - **Regarding the phenomenon:** We would like to clarify that "getting better results after pruning" is **one of our contributions** as discussed in Section 1. We have observed this phenomenon consistently on Mixtral-8x22B, Mixtral-8x7B, Qwen 1.5-MoE-A2.7B, and Qwen2-MoE-A14B. We believe that our evaluation and observation are solid and robust.
>
>   -  **Explanation of such phenomenon:** Indeed, this phenomenon might appear surprising at first glance. However, we discuss the underlying reason in the Section 5.6 of the original submission, and such a phenomenon aligns with the findings in prior work. The reasons are two-fold.
>
>       -  **Why there are pruning patterns better than the full model:** specifically, in Section 5.6, we have discussed that **poorly-trained router network** is the main reason for the performance improvement, which has been observed by many previous works [1][2][3]. In Section 5.6, we verify it by a simple experiment: we select sub-optimal experts based on the output of the router network and show that it can achieve much better performance. This indicates that the router network is unable to select the best experts on downstream tasks. In this case, pruning may have the potential to help the router network eliminate suboptimal choices. Additionally, we demonstrate the significant difference in the expert activate patterns before and after expert pruning, which implies that the performance gain potentially comes from the change in the router network's behavior. These insights align with the findings in prior work [1][2][3].
>       -  **Why such patterns can be found by EEP:** Unlike LLMs' pre-training loss which is not directly related to the metrics that downstream tasks care about, EEP optimizes directly towards the target metric on the specific task, making it easier to fully leverage the information to achieve better results.
>
>       We hope that the above explanation can help address your concerns about such a phenomenon.
>
> - **Small Datasets:** We would like to point out that most datasets we use in Tables 1,3, and 4 are not very small, with more than 500 data points (MultiRC, WIC, BoolQ, ReCoRD, DROP, SQuAD). For datasets like SQuAD and BoolQ, the dataset size is 1000. Especially, the datasets we use in Tables 2, 11, and 13 are the same as NAEE [4] that the reviewer mentioned.
>
> - **MMLU**. We have used MMLU for the OOD pruning setting in our initial submission. Specifically, we run EEP on a subset of it to find candidates with good performance and use another OOD subset for validation. The results are demonstrated in Tables 5, 12, and 13 in our paper.
>
> -  To further address your question, we conduct an additional experiment on MMLU. We run EEP on a subset of MMLU, and test the results on another non-overlapping subset of MMLU. This can be viewed as the task-specific performance on MMLU. The results are shown in the table below. We can see that: **(1)** EEP outperforms baseline methods; and **(2)** When we prune the number of experts from 8 to 6, EEP can even outperform the original full model. These takeaway  messages are consistent with the main results in the paper.
>
>     | Experts | Method     | Accuracy |
>     |---------|------------|----------|
>     | Num=8   | Full Model | 60.7     |
>     | Num=6   | Frequency  | 54.3     |
>     | Num=6   | NAEE       | 57.5     |
>     | Num=6   | EEP        | 61.8     |
>     | Num=4   | Frequency  | 46.7     |
>     | Num=4   | NAEE       | 53.5     |
>     | Num=4   | EEP        | 56.9     |
>
>     [1] Zhou et al., Mixture-of-Experts with Expert Choice Routing
>
>     [2] Fedus et al., Scaling to trillion parameter models with simple and efficient sparsity
>
>     [3] Chi et al., On the representation collapse of sparse mixture of expert.
>
>     [4] Lu et al., Not all experts are equal: Efficient expert pruning and skipping for mixture-of-experts large language models.

---

> ### Author Response · Authors · 2024-11-27
> **Response to Reviewer oSj6 2**
>
> **W2:** In Table 6, even after pruning from 8 experts down to 4 and then 2, the speedup is small. What factors might explain this limited increase in speed?
>
> -  Although the total number of experts is pruned from 8 to 2, there are still two experts activated during the inference, so there is no reduction in the total amount of computation. In this case:
>
>     -  **(1)** If the batch size is large, then the inference process will be a computation-intensive task, meaning that the computation (instead of weight/cache loading) takes up the majority of the time. Therefore, pruning the total number of experts should *not* improve the inference speed too much. To accelerate inference speed in a computation-intensive case further, we need to reduce the number of active experts per token, as shown in "Active=1" rows in Table 6.
>
>     -  **(2)** On the other hand, if the batch size is small, then the inference process will be a memory-access-intensive task, where the inference time is closely related to the parameter size of the model. Therefore, the acceleration ratio will be much larger.
>
> -  For detailed profiling results under different batch sizes, please refer to Table 17 in the paper.
>
> **W3:** The method’s efficiency is largely limited by its computational demands and the dataset used. The pruning dataset is relatively small, yet the process still requires significant time—up to several hours.
>
> - Though EEP is not designed for quick pruning, we would like to point out that the search cost of EEP can be easily reduced by using a smaller size of training sets---we have shown such results in Table  15 of our initial submission. To better illustrate the point,
> we provide more results below, including the performance and the search cost of EEP under 200, 100, and 50 data points. We can see that EEP (Prune only) with 50 data points can complete the pruning process in a very short time (**less than 20 minutes**) while having good performance. Such computational cost is comparable with the baseline method NAEE, but our EEP shows significant performance improvement compared to both the full model and the baseline.
>
>     **Performance of EEP with Different Data Sizes**
>     |                        | WIC  | RTE  | BoolQ | CB   | DROP | SQuAD | Avg   |
>     |------------------------|------|------|-------|------|------|-------|-------|
>     | Original (Prune only)  | 57.8 | 74.0 | 82.8  | 69.6 | 37.3 | 75.2  | 66.1  |
>     | Original (Prune+Merge) | 65.0 | 76.9 | 85.8  | 75.0 | 38.7 | 78.6  | 70.5  |
>     | 200 data (Prune only)   | 58.0 | 71.5 | 82.6  | 69.6 | 36.0 | 76.0  | 65.6- |
>     | 200 data (Prune+Merge)  | 63.4 | 74.3 | 85.0  | 75.0 | 38.1 | 79.2  | 69.2  |
>     | 100 data (Prune only)   | 58.4 | 71.5 | 82.2  | 69.6 | 37.1 | 76.2  | 65.8  |
>     | 100 data (Prune+Merge) | 62.0 | 73.6 | 84.8  | 75.0 | 38.7 | 78.6  | 68.9  |
>     | 50 data (Prune only)    | 56.4 | 68.6 | 78.0  | 76.8 | 34.9 | 77.6  | 65.4  |
>     | 50 data (Prune+Merge)  | 59.4 | 72.6 | 85.2  | 83.9 | 37.9 | 79.6  | 69.8  |
>
>     **Runing Time of EEP and NAEE** (See Table 15 in the revision for full results)
>     |                 | WIC   | RTE   | BoolQ | CB    | DROP  | SQuAD |
>     |-----------------|-------|-------|-------|-------|-------|-------|
>     | NAEE            | 0.11h | 0.13h | 0.19h | 0.15h | 0.26h | 0.21h |
>     | EEP(Prune only) | 0.22h | 0.23h | 0.26h | 0.24h | 0.33h | 0.27h |
>
> - Additionally, in practice, the computational costs during deployment are often much higher than those during training. For example, OpenAI spent more than \\$ 4 billion this year to run the inference workload of ChatGPT, while \$ 3 billion was spent to complete the full training of ChatGPT as well as other models (source: https://www.datacenterdynamics.com/en/news/openai-training-and-inference-costs-could-reach-7bn-for-2024-ai-startup-set-to-lose-5bn-report/).
> Therefore, we focused our evaluation on demonstrating how EEP improves the model efficiency (including GPU memory consumption and inference latency) during deployment, rather than its one-time search cost.
>
> -  We also want to emphasize that one main strength in the computational efficiency of EEP as a gradient-free method is that EEP can be conducted on devices that are only available for inference. In contrast, gradient-based approaches require substantially more GPU memory, which may be an important problem if the user has a limited GPU resources.

---

> ### Author Response · Authors · 2024-11-27
>
> **W4:** Are the training and evaluation samples drawn from the same distribution? This could make sense as a task-agnostic pruning approach. However, given the long pruning time even with limited samples, why not fine-tune a roughly pruned model instead?
>
> -  **Our Pruning Setting:** In task-specific scenarios (results are shown in Tables 1, 2, 3, 4, 9, 10, and 11), training and evaluation samples are drawn from the same distribution. In task-agnostic scenarios (relative are shown in Tables 5, 12, and 13), training and evaluation samples are drawn from different distributions. Experimental results have shown the effectiveness of EEP under both settings.
>
> -  **Comparison with Gradient-based Method:** Gradient-based tuning requires substantially more GPU memory, while EEP can be conducted on devices that can only afford inference. Therefore, our method can be applied to a broader range of applications.
>
> -  To further address the concern, we discuss the relationship between our EEP and gradient-based tuning below and conduct experiments of tuning post-pruning models.
>
>     - **The Relationship between EEP and Gradient-based Methods:** We would like to clarify that EEP can *not* be fully replaced by gradient-based methods. Specifically, phase 1 (expert pruning) is a discrete pruning process that cannot be conducted by fine-tuning; gradient-based methods are only an alternative for phase 2 (expert merging) of EEP. Thus, we only need to consider the comparison between phase 2 (expert merging) and the gradient-based method. The efficiency and effectiveness of phase (expert pruning) 1 of EEP still hold.
>
>     - **Comparison between Model Merging and Fine-tuning:** We conduct experiments by using fine-tuning to mitigate the loss introduced by pruning. We chose the discrete pruning pattern with 4 experts gotten by EEP and NAEE as the initial model to be tuned. For fine-tuning, we use LoRA since the memory cost of full model fine-tuning is too large. For model merging, we use a dataset with a size 50, while for fine-tuning we use the whole training set. Performance and training costs are shown below:
>
>         **Performance of Fine-tuning and Model Merging**
>         |           | WIC  | RTE  | BOOLQ | SQUAD | Drop | Avg  |
>         |-----------|------|------|-------|-------|------|------|
>         | EEP       | 56.4 | 68.6 | 78.0  | 77.6  | 34.9 | 63.1 |
>         | EEP+merge | 59.4 | 72.6 | 85.2  | 79.6  | 37.9 | 66.9 |
>         | EEP+tune  | 63.6 | 69.6 | 83.4  | 79.2  | 37.7 | 66.7 |
>         | NAEE      | 52.6 | 54.5 | 77.0  | 53.0  | 31.2 | 53.7 |
>         | NAEE+tune | 54.6 | 56.0 | 81.4  | 64.1  | 33.3 | 56.6 |
>
>         **Additional Time and Memory Cost of Fine-tuning and Model Merging**
>         |      | Method | WIC   | RTE   | BOOLQ | SQUAD | Drop  |
>         |------|--------|-------|-------|-------|-------|-------|
>         | Time | merge  | 40min | 40min | 45min | 51min | 53min |
>         | Time | tune   | 5min  | 10min | 14min | 10min | 42min |
>         | Mem  | merge  | 4.1G  | 7.2G  | 10G   | 10G   | 6G    |
>         | Mem  | tune   | 19G   | 42G   | 62G   | 56G   | 42G   |
>
>    -  The key takeaways are: **(1) The effectiveness of model merging:** On average, the performance of EEP+merge is slightly better than EEP+fine-tune. **(2) Lower memory cost:** Though the optimization time of model merging is larger than fine-tuning, its additional GPU memory cost is much smaller. Since the memory of the pruned model is 46.6G, it is difficult for users to fine-tune it on a single 80G GPU. **(3) The effectiveness of EEP discrete pruning:** By comparing the third row and fifth row, we can see that EEP discrete pruning is a much better initialization for tuning than the baseline method, which is also a key contribution of this work.

---

> ### Author Response · Authors · 2024-12-01
> **Follow-up Comment**
>
> Thank you for your insightful review. We sincerely appreciate your recognition of the clarity, novelty and effectiveness of EEP.
>
> Since your feedback has been crucial in refining our work, we have worked diligently in our rebuttal to address the concerns and questions raised. If you find that our response has satisfactorily resolved your concerns, we would kindly ask you to consider revisiting your score to better reflect your updated evaluation of the paper.
>
> Thank you again for your time and insightful review. If you have any further questions, we are very happy to discuss.

---

> > ### Comment · Reviewer_oSj6 · 2024-12-01
> >
> > Thanks for your efforts during the discussion. NAEE caches all the input and output of each expert during one forward pass and conducts the enumeration process with a fixed number of iterations to search for the best candidates to prune. However, I notice that the reported running time of your results varies a lot across different datasets. Why does this happen?
> >
> > Besides, is there any curve to show how the model performance changes across generations? Or how the expert or weight is chosen dynamically. In my own understanding, the evolutional strategy is largely impacted by randomness. I wonder whether there is a pattern for learning to prune and how is the statistic significance.

---

> ### Author Response · Authors · 2024-12-02
> **Follow-up Response**
>
> Thank you for your response. Here are our answers to your questions.
>
> **Q1: Different Running time of NAEE on Different Datasets.**
>
> - Thank you for your question. We first fix a typo of the running time of NAEE on SQuAD dataset in Table "Runing Time of EEP and NAEE" in our previous rebuttal titled "Response to Reviewer oSj6 2". Then we will discuss our implementation and the reason of different running time below.
>
> - **Our implementation of NAEE.** Since we need to perform pruning for question-answer tasks, the input data we use for pruning should ideally include both the questions and the answers. Therefore, we first run the entire generation process with the full model and cache the intermediate features. Then we enumerate all possible pruning patterns and calculate the loss with results from the full model for each layer. We finally select pruning patterns with minimum loss.
>
> -  We think the main reason different running time is the **different sequence length.** For each dataset, the average sequence length of the prompt (i.e., the question and the default in-context learning content of OpenCompass) and generation length are both different. We provide the average token length of each dataset below. We can see that the token length is positively correlated with the running time.
>
>     | Dataset       | WIC | RTE | BoolQ | CB  | DROP | SQuAD |
>     |---------------|-----|-----|-------|-----|------|-------|
>     | Token Length  | 105 | 258 | 430   | 340 | 1036 | 467   |
>
> **Q2.1: Any curve to show how the model performance changes across generations.**
>
> We have already included several performance-search iteration curves in Figures 6, 7, and 8 of our revision. Please refer to these figures for further details.
>
> **Q2.2: the impact of randomness on search.**
>
> To address your concern and show the robust of EEP to randomness, we further conduct experiments using different random seeds. We report the results of the pruning phase using 50 data under 3 random seeds below.
>
> |                   | BoolQ         | WIC           |
> |-------------------|---------------|---------------|
> | Seed1 (Oringinal) | 78.0          | 56.4          |
> | Seed2             | 82.8          | 60.8          |
> | Seed3             | 82.8          | 58.2          |
> | Avg.              | 81.2$\pm$2.26 | 58.5$\pm$1.81 |
>
> **Q2.3: how expert change dynamically & pattern for learning to prune**
>
> We demonstrate the changing pattern of both pruning phase and expert merging phase in https://anonymous.4open.science/r/EEP_pattern_ano-F708. Since we divided all layers into 4 groups for these datasets, we demonstrate the pattern of all groups. For the discrete pruning phase, we use dark-colored blocks to display the expert selected in each iteration. For the expert merging phase, we display the change curves of all 4x8 coefficients (4 is the total number of kept experts and 8 is the number of all experts before pruning) for each group. As we can see, search patterns change at the beginning and tend to stabilize after few iterations. We note that although evolutionary strategies relies on noise to explore the optimal solution, it can also converge to a stable solution because the best candidate selected for crossover and evaluation depends on the ranking with respect to fitness.

---

> > ### Comment · Reviewer_oSj6 · 2024-12-02
> >
> > Thank you for your response. Based on the following reasons, I will maintain my overall score:
> >
> > 1. The experiments are conducted on a small dataset, including the out-of-distribution (OOD) sub-MMLU dataset, which makes the results unreliable and unclear in terms of real-world performance (the pruned model even outperforms the original full model). Additionally, some baseline methods perform worse than a random strategy, which raises further concerns about the validity of the results.
> >
> > 2. Considering the significant time consumption of the evolutionary strategy, the improvements are minor and lack scalability to broader tasks. Although the method proposed in this paper introduces a more fine-grained weight search and merging strategy between different experts, the mixing of rows and columns disrupts the original learning space. In other words, this creates a lot of unusable search space, leading to inefficiency in the search process. It would be beneficial to add stronger regularization to the search process to improve its efficiency.
> >
> > For these reasons, I am not fully confident in the paper’s conclusions. I will maintain the overall score but reduce my confidence in it.

---

> ### Author Response · Authors · 2024-12-02
> **Follow-up Response 2**
>
> Thank you for your response. We **sincerely suggest you reading our rebuttal (including this response) and our revision carefully** regarding your concerns. We would like to provide further clarification on your question here.
>
> > Q1.1 Small Dataset
>
> We have already answered this question in our original rebuttal titled "Response to Reviewer oSj6 1". Many datasets in Tables 1,3 and 4 are not small, with more than 500 data points independently and identically sampled from the entire dataset. The datasets used in Tables 2, 11 and 13 are the same as NAEE. For the OOD pruning setting, we validate our pruning pattern on **all these datasets, which contains several thousands test data in total and is far from being a small validation dataset**, and outperforming the baseline method significantly.
>
> > Q1.2 Better than the full model
>
> We have already discussed the reasons why EEP can outperform the full model in a task-specific setting in Section 5.6. Here, we would like to reiterate them: **(1)** As demonstrated in previous works [1][2][3], the router network is not well trained and may not be suitable for downstream tasks. **(2)** EEP can optimize the metrics directly, which provides better signals for the optimization. We have conducted **extensive experiments** to demonstrate that outperforming the full model on specific tasks is indeed feasible. Reviewer **CAN** reproduce the results with the original checkpoint and the sparse pattern provided in our code in the supplementary material.
>
> [1] Zhou et al., Mixture-of-Experts with Expert Choice Routing
>
> [2] Fedus et al., Scaling to trillion parameter models with simple and efficient sparsity
>
> [3] Chi et al., On the representation collapse of sparse mixture of expert.
>
> > Q1.3 Some baseline methods perform worse than a random strategy
>
> For frequency and soft-frequency baselines, it is reasonable that the they perform poorly. As discussed in Section 5.6, the router network does not train well on downstream tasks, making it unreasonable to select pruning patterns based on the output of the router network. For the NAEE baseline, it outperforms random pruning in most cases. For the only exception on SQuAD dataset in Table 1, **we have already provided code in our supplementary material to reproduce the random pruning results**. Additionally, as demonstrated in Table 7, the router network performs badly on SQuAD dataset and the performance of the full model is suboptimal. Therefore, trying to keep consistent with the full model may not be a good choice for pruning on this dataset and it is reasonable that random pruning outperforms the full model and the baseline.
>
> > Q2.1 significant time consumption of the evolutionary strategy
>
> We have already shown in the table "Performance of EEP with Different Data Sizes" and "Runing Time of EEP and NAEE" of our previous rebuttal titled "Response to Reviewer oSj6 2" that only conducting the pruning phase of EEP with reduced dataset size is much more efficient and also effective. Specifically, the performance of this method can **still outperform the full model and all baselines**, while can be done in **less than 20 minutes**. We neither consider completing the pruning within 20 minutes as "significant time consumption", nor do we believe there is a substantial difference between finishing in 20 minutes and 10 minutes as the baseline methods do on a downstream dataset. Additionally, as we discussed in our previous rebuttal titled "Response to Reviewer oSj6 2", we believe that the training cost is much smaller than the inference cost. Our EEP mainly focus on efficient inference. And as mentioned earlier, its current optimization is already highly efficient.

---

> > ### Author Response · Authors · 2024-12-02
> > **Follow-up Response 3**
> >
> > > Q2.2 ...the mixing of rows and columns disrupts the original learning space. EEP creates a lot of unusable search space, leading to inefficiency in the search process...
> >
> > - **Regarding the inefficiency of EEP** We want to point out that the fact is that, the search results of model merging phase has already led to significant improvements. As an independent option, if the users are unwilling to incur additional costs, they can choose to perform only the pruning phase or conduct fewer search iterations of the model merging phase, which is already effective and efficient enough as we discussed before. Model merging provides a further chance to enhance the performance.
> >
> > - **Regarding the mixing of rows and columns** We are not fully understand what you mean by "mixing of rows and columns". We want to point out that **the essence of the expert merging phase is to search for the coefficients of other experts being merged in**. The matrix we introduced is just a convenient way to express this process. If you mean conducting expert merging itself, we would like to further point out that there have already been lots of works illustrating the effectiveness of model merging method [1][2][3][4]. Among these papers, [1] merges different weight inside one model, which is similar to EEP. We do not think the conducting expert merging will disrupts the original learning space.
> >
> >   [1] Akiba et al., Evolutionary Optimization of Model Merging Recipes, Arxiv.
> >
> >   [2] Jin et al., Dataless Knowledge Fusion By Merging Weights Of Language Models, ICLR 2023
> >
> >   [3] Matena \& Raffel, Merging Models with Fisher-Weighted Averaging, NeurIPS 2022
> >
> >   [4] Liu et al., Checkpoint Merging via Bayesian Optimization in LLM Pretraining, Arxiv.
> >
> > - We thank your suggestion of adding stronger regularization. However, we think the results of EEP are already good enough to support our conclusion.
> >
> > We thank you again for your time and effort in reviewing the paper.

---

### Official Review · Reviewer_mRrF · 2024-11-01

**Soundness:** 3
**Presentation:** 3
**Contribution:** 3
**Rating:** 6
**Confidence:** 4

**Summary:**

This paper introduces Efficient Expert Pruning (EEP), a gradient-free evolutionary strategy for optimizing Sparse Mixture-of-Experts (SMoE) language models. The work addresses the challenge of reducing model size and computation costs while maintaining performance. The authors propose a two-phase approach combining expert pruning and expert merging, which can be executed using only model inference capabilities without requiring gradient computation or extensive GPU resources. The method is implemented through router mapping and expert merging matrices, enabling evolutionary search for optimal pruning configurations.

**Strengths:**

1. The paper is exceptionally well-written and organized, with comprehensive experimental validation across multiple model scales and diverse datasets. The methodology is clearly presented with thorough ablation studies and reproducible code.
2. The proposed two-phase EEP framework presents a novel and effective paradigm for model compression. The authors provide insightful analysis of router network behavior that justifies their approach.
3. The method achieves remarkable results while only requiring inference capabilities, with pruned models sometimes even outperforming the original uncompressed models.

**Weaknesses:**

1. The analysis of why performance sometimes improves after pruning in Sec. 5.6. remains somewhat speculative, lacking rigorous theoretical foundations and comprehensive empirical validation.
2. The performance gap between global setting and task-specific setting in OOD experiments suggests that EEP's impressive results might be more dependent on task-specific optimization than claimed, potentially limiting its general applicability across different tasks.
3. The search process of EEP is computationally expensive and time-consuming, particularly for large datasets. The paper does not address this efficiency bottleneck.

**Questions:**

Could the authors elaborate on how the choice of group numbers and search iterations affects the final performance? A more systematic analysis of these hyperparameters would be valuable for implementation.

---

> ### Author Response · Authors · 2024-11-27
> **Response to Reviewer mRrF 1**
>
> We greatly appreciate the reviewer’s constructive feedback and insightful suggestions, which have been instrumental in helping us enhance our paper. Below, we address your questions one-by-one.
>
> **W1:** The analysis of why performance sometimes improves after pruning in Sec. 5.6. remains somewhat speculative, lacking rigorous theoretical foundations and comprehensive empirical validation.
>
> -  In Section 5.6, we have discussed that why our methods can sometimes improve the performance. The reasons are two-fold.
>
>   -  **Why there are pruning patterns better than the full model even without parameter update:** We think **poorly-trained router network** is the main reason for the performance improvement, which has been observed by many previous works [1][2][3]. In Section 5.6, we verify it by a simple experiment: we select sub-optimal experts based on the output of the router network and show that it can achieve much better performance. This indicates that the router network is unable to select the best experts on downstream tasks. In this case, pruning may have the potential to help the router network eliminate suboptimal choices. Additionally, we demonstrate the significant difference in the expert activate patterns before and after expert pruning, which implies that the performance gain potentially comes from the change in the router network's behavior. These insights align with the findings in prior work [1][2][3].
>
>   -  **Why such patterns can be found by EEP:** Unlike LLMs' pre-training loss which is not directly related to the metrics that downstream tasks care about, EEP optimizes directly towards the target metric on the specific task, making it easier to fully leverage the information to achieve better results.
>
> -  We would like to point out that understanding the router network's behavior is an important research direction, however, we believe a comprehensive study needs specific investigation and effort. Since our paper proposes a practical pruning method, we mainly focus on evaluating our method with extensive experiments, and we reserve a more theoretical exploration for future work.
>
> - If the reviewer has other concerns or suggestions regarding this part, please let us know and we are happy to discuss further!
>
>     [1] Zhou et al., Mixture-of-Experts with Expert Choice Routing
>
>     [2] Fedus et al., Scaling to trillion parameter models with simple and efficient sparsity
>
>     [3] Chi et al., On the representation collapse of sparse mixture of expert.
>
> **W2:** The performance gap between global setting and task-specific setting in OOD experiments suggests that EEP's impressive results might be more dependent on task-specific optimization than claimed, potentially limiting its general applicability across different tasks.
>
> -  We agree EEP can achieve more impressive results on task-specific optimization scenarios. We would like to point out two facts regarding this:
>
>     - Task-specific pruning also has a lot of application scenarios. Task-specific LLMs fine-tuned from general capability LLMs, such as Code LLama, BioMistral, and Llama Guard, are widely used in many applications. Recent papers [1] also argue that composing different task-specific models/modules rather than using one general-purpose model and expecting it to perform well on all tasks might be better in terms of scalability, efficiency, and so on. Using EEP to adapt a general-purpose MoE model towards one specific downstream task while making it more efficient aligns with these trends.
>
>     - As shown in Table 15, EEP can complete the pruning process in **less than 20 minutes** (if pruned only) while achieving better performance compared to the full model and baselines. It indicates that the adaptation of EEP to downstream tasks is relatively quick, making it a practical algorithm for task-specific optimization.
>
> -  Additionally, in the task-agnostic scenario, although EEP does not achieve significant gain after pruning compared with the original model, the experimental results still show the superior performance of EEP compared with other pruning methods, which demonstrates the value of our method.
>
> [1] Xiao et al. "Configurable foundation models: Building llms from a modular perspective."

---

> ### Author Response · Authors · 2024-11-27
> **Response to Reviewer mRrF 2**
>
> **W3:** The search process of EEP is computationally expensive and time-consuming, particularly for large datasets. The paper does not address this efficiency bottleneck.
>
> - Though EEP is not designed for quick pruning, we would like to point out that the search cost of EEP can be easily reduced by using a smaller size of training sets---we have shown such results in Table  15 of our initial submission. To better illustrate the point, we provide more results below, including the performance and the search cost of EEP under 200, 100, and 50 data points. We can see that EEP (Prune only) with 50 data points can complete the pruning process in a very short time (**less than 20 minutes**) while having good performance.  Such computational cost is comparable with the baseline method NAEE, but our EEP shows significant performance improvement compared to both the full model and the baseline.
>
>     **Performance of EEP with Different Data Sizes**
>     |                        | WIC  | RTE  | BoolQ | CB   | DROP | SQuAD | Avg   |
>     |------------------------|------|------|-------|------|------|-------|-------|
>     | Original (Prune only)  | 57.8 | 74.0 | 82.8  | 69.6 | 37.3 | 75.2  | 66.1  |
>     | Original (Prune+Merge) | 65.0 | 76.9 | 85.8  | 75.0 | 38.7 | 78.6  | 70.5  |
>     | 200 data (Prune only)   | 58.0 | 71.5 | 82.6  | 69.6 | 36.0 | 76.0  | 65.6- |
>     | 200 data (Prune+Merge)  | 63.4 | 74.3 | 85.0  | 75.0 | 38.1 | 79.2  | 69.2  |
>     | 100 data (Prune only)   | 58.4 | 71.5 | 82.2  | 69.6 | 37.1 | 76.2  | 65.8  |
>     | 100 data (Prune+Merge) | 62.0 | 73.6 | 84.8  | 75.0 | 38.7 | 78.6  | 68.9  |
>     | 50 data (Prune only)    | 56.4 | 68.6 | 78.0  | 76.8 | 34.9 | 77.6  | 65.4  |
>     | 50 data (Prune+Merge)  | 59.4 | 72.6 | 85.2  | 83.9 | 37.9 | 79.6  | 69.8  |
>
>      **Running Time of EEP (50 data) and NAEE** (See Table 15 in the revision for full results)
>     |                 | WIC   | RTE   | BoolQ | CB    | DROP  | SQuAD |
>     |-----------------|-------|-------|-------|-------|-------|-------|
>     | NAEE            | 0.11h | 0.13h | 0.19h | 0.15h | 0.26h | 0.21h |
>     | EEP(Prune only) | 0.22h | 0.23h | 0.26h | 0.24h | 0.33h | 0.27h |
>
> -  Additionally, in practice, the computational costs during deployment are often much higher than those during training. For example, OpenAI spent more than 4 billion dollars this year to run the inference workload of ChatGPT, while \$ 3 billion was spent to complete the full training of ChatGPT as well as other models (source: https://www.datacenterdynamics.com/en/news/openai-training-and-inference-costs-could-reach-7bn-for-2024-ai-startup-set-to-lose-5bn-report/). Therefore, we focused our evaluation on demonstrating how EEP improves the model efficiency (including GPU memory consumption and inference latency) during deployment, rather than its one-time search cost.
>
> -  We also want to emphasize that one main strength in the computational efficiency of EEP as a gradient-free method is that EEP can be conducted on devices that are only available for inference. In contrast, gradient-based approaches require substantially more GPU memory, which may be an important problem if the user has a limited GPU resources.

---

> ### Author Response · Authors · 2024-11-27
> **Response to Reviewer mRrF 3**
>
> **Q1:** Could the authors elaborate on how the choice of group numbers and search iterations affects the final performance? A more systematic analysis of these hyperparameters would be valuable for implementation.
>
> -  In the paper, we have already provided ablation studies of these two factors; see Appendix D.8. To alleviate the concern, we report results using more groups and add more curves on the performance vs. search iterations on other datasets. Results are provided in Table 21 and Figures 6, 7, 8. The table of different numbers of groups is copied below. If reviewers want to see specific experimental results, we are also glad to include them.
>
> -  The observation remains consistent with our previous results: **(1)** increasing the number of groups is very likely to bring better results due to the expanded optimization space, and **(2)** evolutionary search can already find satisfying results in tens of iterations.
>
>     | Group Number | Num Expert | Method      | RTE  | DROP | ReCoRD |
>     |--------------|------------|-------------|------|------|--------|
>     | 4            | 4          | Prune only  | 62.8 | 35.5 | 59.2   |
>     | 4            | 4          | Prune+Merge | 71.5 | 38.9 | 63.2   |
>     | 4            | 2          | Prune only  | 53.8 | 25.3 | 36.0   |
>     | 4            | 2          | Prune+Merge | 61.7 | 27.5 | 38.8   |
>     | 8            | 4          | Prune only  | 71.1 | 33.1 | 58.8   |
>     | 8            | 4          | Prune+Merge | 71.1 | 38.1 | 59.2   |
>     | 8            | 2          | Prune only  | 54.5 | 26.7 | 36.8   |
>     | 8            | 2          | Prune+Merge | 57.4 | 26.7 | 40.8   |
>     | 16           | 4          | Prune only  | 72.9 | 33.3 | 58.4   |
>     | 16           | 4          | Prune+Merge | 75.8 | 35.2 | 58.4   |
>     | 16           | 2          | Prune only  | 57.4 | 27.5 | 41.2   |
>     | 16           | 2          | Prune+Merge | 58.8 | 30.4 | 43.2   |
>     | 32           | 4          | Prune only  | 74.0 | 37.3 | 60.0   |
>     | 32           | 4          | Prune+Merge | 76.9 | 39.7 | 63.6   |
>     | 32           | 2          | Prune only  | 64.3 | 37.1 | 47.2   |
>     | 32           | 2          | Prune+Merge | 69.0 | 38.4 | 47.2   |

---

> ### Author Response · Authors · 2024-12-01
> **Follow-up Comment**
>
> Thank you for your insightful review and your initial positive evaluation of our work. We sincerely appreciate your recognition of the novelty and effectiveness of EEP, as well as the comprehensive experiments.
>
> Since your feedback has been crucial in refining our work, we have worked diligently in our rebuttal to address the concerns and questions raised. We hope our responses and additional clarifications have further strengthened your confidence in the contributions of our work.
>
> If you find that our response has satisfactorily resolved your concerns, we would kindly ask you to consider revisiting your score to better reflect your updated evaluation of the paper. We respect your discretion and remain grateful for your valuable feedback and initial positive evaluation.
>
> Thank you again for your time and insightful review. If you have any further questions, we are very happy to discuss.

---

> > ### Comment · Reviewer_mRrF · 2024-12-03
> >
> > Thank you for your comprehensive response. I maintain my original score.

---

### Official Review · Reviewer_QTE1 · 2024-11-06

**Soundness:** 2
**Presentation:** 2
**Contribution:** 2
**Rating:** 6
**Confidence:** 4

**Summary:**

This paper works on the topic of pruning experts in SMoE models. They propose a gradient-free evolutionary strategy named EEP. In EEP, there are two matrices named expert merging matrix and experts mapping matrix, used to search for pruning configurations. The goal is to achieve sparsity while reaching good performance on downstream tasks. The paper shows the performance of the proposed method on four different model sizes and uses more than 10 datasets.

**Strengths:**

SMoE can achieve faster inference while maintaining performance. To search for the subset of experts, this paper proposes a gradient-free approach based on evolutionary strategy. By using two phases expert pruning and expert merging, the method achieves high sparsity.

**Weaknesses:**

1. While the method is gradient-free, it does not necessarily imply a low search cost. The evolutionary strategy used requires multiple iterations and epochs, and the search cost is highly dependent on the settings for iterations, epochs, and the data size used for calculating the validation metric. To better illustrate the efficiency of the method, I suggest the authors provide concrete runtime numbers for different dataset sizes, comparing them to gradient-based alternatives. This would help clarify whether the method can be effectively extended to more complex datasets.

2. The method heavily relies on evaluation results from selected data to conduct the search process. While Table 15 shows in-distribution results for two dataset sizes, it is unclear whether similar trends hold for out-of-distribution cases. A comprehensive study on the influence of dataset size is recommended. Specifically, it would be beneficial to include a plot showing the performance vs. dataset size across a range of sizes for both in-distribution and out-of-distribution settings. Additionally, the authors should explain the counterintuitive result where a smaller dataset size provides better performance than a larger dataset size.

3. For out-of-distribution evaluation, 7 datasets from MMLU are used for validation. It would be helpful to know whether the number of validation datasets affects performance. To address this, I recommend conducting an ablation study that varies the number of validation datasets (e.g., from 1 to 7) and reporting how this impacts performance. This would offer insight into the method’s robustness in extreme cases where only a single dataset is available for validation.

4. The writing of the paper could be improved, as the key ideas are not easy to follow. A significant portion of the text is devoted to background and preliminaries, such as the description of the self-attention mechanism in Section 3, which is well-known and could be condensed or moved to the appendix. This would create more space to emphasize the main algorithm and core contributions in the main paper. I suggest reorganizing the structure to focus on the core algorithm earlier in the text for better clarity and accessibility.

**Questions:**

1. Why is the full model performance significantly higher in this paper compared to the NAEE paper? For example, in Table 2, the full model (Mixtral 8×7B-Instruct) achieves 85.8 on ARC-c with 8 experts, while in the NAEE paper's Table 1, the performance is only 62.20.

2. Can the proposed method still work for out-of-distribution cases when there is only one dataset available for validation?

3. What does the performance vs. dataset size curve look like during the search process? What is the rationale for choosing specific dataset sizes for validation?

4. How is the performance of the proposed method compared to dynamic MoE methods?

---

> ### Author Response · Authors · 2024-11-27
> **Response to Reviewer QTE1 1**
>
> We greatly appreciate the reviewer’s constructive feedback and insightful suggestions, which have been instrumental in helping us enhance our paper. Below, we address your questions one-by-one.
>
> **W1:** While the method is gradient-free, it does not necessarily imply a low search cost...
>
> -  We want to clarify that we did not argue for low search costs in this paper.
> What we argued is that our gradient-free method can be conducted on devices that are only available for inference (see lines 92-93 in our original submission). In contrast, gradient-based approaches require substantially more GPU memory, which may be an important problem if the user has a limited GPU resources.
>
> -  Additionally, we think the search cost is not a main drawback for our EEP due to the following reasons. **(1)** Though EEP is not designed for quick pruning, we would like to point out that the search cost of EEP can be easily reduced by using a smaller size of training sets---**we have shown "concrete runtime numbers for different dataset sizes" in Table 15 of our initial submission**. To better illustrate the point, we provide more results below, including the performance of EEP under 200, 100, and 50 data points and the search cost of 50 data (More results can be found in the Table 15 of the revision). We can see that EEP (Prune only) with 50 data points can complete the pruning process in a very short time (**less than 20 minutes**) while having good performance.  Such computational cost is comparable with the baseline method NAEE, but our EEP shows significant performance improvement compared to both the full model and the baseline. **(2)** Additionally, in practice, the computational costs during deployment are often much higher than those during training. For example, OpenAI spent more than 4 billion dollars this year to run the inference workload of ChatGPT, while \$ 3 billion was spent to complete the full training of ChatGPT as well as other models (source: https://www.datacenterdynamics.com/en/news/openai-training-and-inference-costs-could-reach-7bn-for-2024-ai-startup-set-to-lose-5bn-report/). Therefore, we focused our evaluation on demonstrating how EEP improves the model efficiency (including GPU memory consumption and inference latency) during deployment, rather than its one-time search cost.
>
>     |                        | WIC  | RTE  | BoolQ | CB   | DROP | SQuAD | Avg   |
>     |------------------------|------|------|-------|------|------|-------|-------|
>     | Original (Prune only)  | 57.8 | 74.0 | 82.8  | 69.6 | 37.3 | 75.2  | 66.1  |
>     | Original (Prune+Merge) | 65.0 | 76.9 | 85.8  | 75.0 | 38.7 | 78.6  | 70.5  |
>     | 200 data (Prune only)   | 58.0 | 71.5 | 82.6  | 69.6 | 36.0 | 76.0  | 65.6- |
>     | 200 data (Prune+Merge)  | 63.4 | 74.3 | 85.0  | 75.0 | 38.1 | 79.2  | 69.2  |
>     | 100 data (Prune only)   | 58.4 | 71.5 | 82.2  | 69.6 | 37.1 | 76.2  | 65.8  |
>     | 100 data (Prune+Merge) | 62.0 | 73.6 | 84.8  | 75.0 | 38.7 | 78.6  | 68.9  |
>     | 50 data (Prune only)    | 56.4 | 68.6 | 78.0  | 76.8 | 34.9 | 77.6  | 65.4  |
>     | 50 data (Prune+Merge)  | 59.4 | 72.6 | 85.2  | 83.9 | 37.9 | 79.6  | 69.8  |
>
>      **Running Time of EEP (50 data) and NAEE** (See Table 15 in the revision for full results)
>     |                 | WIC   | RTE   | BoolQ | CB    | DROP  | SQuAD |
>     |-----------------|-------|-------|-------|-------|-------|-------|
>     | NAEE            | 0.11h | 0.13h | 0.19h | 0.15h | 0.26h | 0.21h |
>     | EEP(Prune only) | 0.22h | 0.23h | 0.26h | 0.24h | 0.33h | 0.27h |
>
> -  To further address the concern, we discuss the relationship between our EEP and gradient-based tuning below and conduct experiments of tuning post-pruning models.
>
>     -  **The Relationship between EEP and Gradient-based Methods:** We would like to clarify that EEP can *not* be fully replaced by gradient-based methods. Specifically, phase 1 (expert pruning) is a discrete pruning process that cannot be conducted by fine-tuning; gradient-based methods are only an alternative for phase 2 (expert merging) of EEP.  Thus, we only need to consider the comparison between phase 2 (expert merging) and the gradient-based method. The efficiency and effectiveness of phase (expert pruning) 1 of EEP still hold.

---

> ### Author Response · Authors · 2024-11-27
> **Response to Reviewer QTE1 2**
>
> **(continue on Response for W1...)**
> -  To further address the concern...
>
>     - **Comparison between Model Merging and Fine-tuning:** We conduct experiments by using fine-tuning to mitigate the loss introduced by pruning. We chose the discrete pruning pattern with 4 experts gotten by EEP and NAEE as the initial model to be tuned. For fine-tuning, we use LoRA since the memory cost of full model fine-tuning is too large. For model merging, we use a dataset with a size 50, while for fine-tuning we use the whole training set. Performance and training costs are shown below:
>
>         Performance of Fine-tuning and Model Merging
>         |           | WIC  | RTE  | BOOLQ | SQUAD | Drop | Avg  |
>         |-----------|------|------|-------|-------|------|------|
>         | EEP       | 56.4 | 68.6 | 78.0  | 77.6  | 34.9 | 63.1 |
>         | EEP+merge | 59.4 | 72.6 | 85.2  | 79.6  | 37.9 | 66.9 |
>         | EEP+tune  | 63.6 | 69.6 | 83.4  | 79.2  | 37.7 | 66.7 |
>         | NAEE      | 52.6 | 54.5 | 77.0  | 53.0  | 31.2 | 53.7 |
>         | NAEE+tune | 54.6 | 56.0 | 81.4  | 64.1  | 33.3 | 56.6 |
>         -   **Additional Time and Memory Cost of Fine-tuning and Model Merging**
>
>         |      | Method | WIC   | RTE   | BOOLQ | SQUAD | Drop  |
>         |------|--------|-------|-------|-------|-------|-------|
>         | Time | merge  | 40min | 40min | 45min | 51min | 53min |
>         | Time | tune   | 5min  | 10min | 14min | 10min | 42min |
>         | Mem  | merge  | 4.1G  | 7.2G  | 10G   | 10G   | 6G    |
>         | Mem  | tune   | 19G   | 42G   | 62G   | 56G   | 42G   |
>
>     -  The key takeaways are: **(1) The effectiveness of model merging:** On average, the performance of EEP+merge is slightly better than EEP+fine-tune. **(2) Lower memory cost:** Though the optimization time of model merging is larger than fine-tuning, its additional GPU memory cost is much smaller. Since the memory of the pruned model is 46.6G, it is difficult for users to fine-tune it on a single 80G GPU. **(3) The effectiveness of EEP discrete pruning:** By comparing the third row and fifth row, we can see that EEP discrete pruning is a much better initialization for tuning than the baseline method, which is also a key contribution of this work.
>
> **W2 & Q3:** ...it would be beneficial to include a plot showing the performance vs. dataset size across a range of sizes for both in-distribution and out-of-distribution settings. Additionally, the authors should explain the counterintuitive result where a smaller dataset size provides better performance than a larger dataset size \& What does the performance vs. dataset size curve look like during the search process? What is the rationale for choosing specific dataset sizes for validation?
>
> -  For the in-distribution setting, we have already reported some results with smaller dataset sizes in Table 15. More comprehensive results are provided above.
>
> -  For the out-of-distribution setting, we conduct two additional experiments by using only 20 and 35 subsets randomly chosen from the original 50 subsets. The results are shown below.
>
>     |Method| Train set num | Train data num | WIC   | WSC   | RTE   | BoolQ | CB    | DROP  | SQuAD | Avg. |
>     |------|---------------|----------------|-------|-------|-------|-------|-------|-------|-------|------|
>     |EEP| 50            | 773            | 52.4  | 69.2  | 52.0  | 83.2  | 44.6  | 34.4  | 65.2  | 57.3 |
>     |EEP| 35            | 495            | 49.6  | 69.2  | 71.5  | 82.8  | 42.9  | 32.5  | 73.4  | 60.3 |
>     |EEP| 20            | 265            | 50.8  | 67.3  | 67.9  | 84.0  | 41.1  | 33.1  | 62.8  | 58.1 |
>     |NAEE     | 50 | 773            | 54.2  | 60.6  | 55.2  | 69.4  | 53.6  | 30.7  | 45.2  | 52.7 |
>
> - We have already included plots for both settings in Figure 5 of our revision to demonstrate the relationship more clearly.
>
> - From these results, we can find that EEP is very stable with respect to the size of the dataset, since using a much smaller training set will not have a significant impact on the results.
>
> - **Explanation of why smaller dataset size provides better performance:** We would like to point out that in most cases, **the results of different dataset sizes are close to each other**, since EEP is robust to data number. The only exception is CB, where the result of 50-data is much better. We think this is because the test set of CB is very small with only 56 data, while other datasets all have more than several hundreds of data. Therefore, the impact of randomness is much larger on this dataset, leading to the results in Table 15.
>
> - **Rationale for choosing specific dataset sizes for validation:** We do not use validation set in our in-distribution experiments. For out-of-distribution settings, we **randomly** draw a small number (7 in our default setting) of subsets as the validation set that are unseen in the search process.

---

> ### Author Response · Authors · 2024-11-27
> **Response to Reviewer QTE1 3**
>
> **W3 & Q2:** For out-of-distribution evaluation, 7 datasets from MMLU are used for validation. It would be helpful to know whether the number of validation datasets affects performance \& Can the proposed method still work for out-of-distribution cases when there is only one dataset available for validation?
>
> -  Thanks for your suggestions! Based on the above experiments, we further randomly sample some subsets among the 7 validation sets. We randomly choose 4 and 1 as the number of subsets. Additionally, we conduct experiments with 3 random seeds for dataset selection to mitigate the influence of randomness on the conclusions. The results of keeping 6 experts are listed below:
>
>     |Method| Train set Number | Val set Number | Seed | WIC  | WSC  | RTE  | BoolQ | CB   | DROP | SQuAD | Avg. |
>     |----------|------------------|----------------|------|------|------|------|-------|------|------|-------|------|
>     |EEP| 20               | 7              | 1    | 50.8 | 67.3 | 67.9 | 84.0  | 41.1 | 33.1 | 62.8  | 58.1 |
>     |EEP| 20               | 4              | 1    | 50.8 | 67.3 | 67.9 | 84.0  | 41.1 | 33.1 | 62.8  | 58.1 |
>     |EEP| 20               | 4              | 2    | 58.0 | 31.7 | 50.5 | 74.6  | 55.4 | 30.9 | 56.8  | 51.1 |
>     |EEP| 20               | 4              | 3    | 53.8 | 42.3 | 50.2 | 68.4  | 60.7 | 31.7 | 63.6  | 53.0 |
>     |EEP| 20               | 1              | 1    | 50.8 | 67.3 | 67.9 | 84.0  | 41.1 | 33.1 | 62.8  | 58.1 |
>     |EEP| 20               | 1              | 2    | 50.8 | 67.3 | 67.9 | 84.0  | 41.1 | 33.1 | 62.8  | 58.1 |
>     |EEP| 20               | 1              | 3    | 50.8 | 67.3 | 67.9 | 84.0  | 41.1 | 33.1 | 62.8  | 58.1 |
>     |EEP| 35               | 7              | 1    | 49.6 | 69.2 | 71.5 | 82.8  | 42.9 | 32.5 | 73.4  | 60.3 |
>     |EEP| 35               | 4              | 1    | 49.6 | 69.2 | 71.5 | 82.8  | 42.9 | 32.5 | 73.4  | 60.3 |
>     |EEP| 35               | 4              | 2    | 49.6 | 69.2 | 71.5 | 82.8  | 42.9 | 32.5 | 73.4  | 60.3 |
>     |EEP| 35               | 4              | 3    | 51.6 | 69.2 | 66.8 | 83.0  | 41.1 | 35.2 | 65.8  | 59.0 |
>     |EEP| 35               | 1              | 1    | 49.6 | 69.2 | 71.5 | 82.8  | 42.9 | 32.5 | 73.4  | 60.3 |
>     |EEP| 35               | 1              | 2    | 51.6 | 69.2 | 66.8 | 83.0  | 41.1 | 35.2 | 65.8  | 59.0 |
>     |EEP| 35               | 1              | 3    | 49.0 | 63.5 | 52.4 | 75.0  | 46.4 | 32.8 | 59.0  | 54.0 |
>     | NAEE |    50      | -              | -    | 54.2 | 60.6 | 55.2 | 69.4  | 53.6 | 30.7 | 45.2  | 52.7 |
>
> - We can find that even with a limited number (e.g., only 1) of the validation set, EEP can still perform similarly to the results on the entire validation set. More importantly, these results also outperform the baseline methods.
>
> - We also want to point out that, in practice, if the goal is to have good out-of-distribution performance, it makes more sense to use search/validation sets that are as large as possible. Artificially reducing the dataset size offers no benefits. Therefore, in our original submission, we chose to use the whole MMLU, a large, diverse, and public dataset, to conduct the experiments, which we think is a more realistic setting.
>
> **W4:** The writing of the paper could be improved, as the key ideas are not easy to follow...
>
> -  We appreciate the reviewer’s suggestion to improve the writing. The background section (Section 3) was designed to provide the necessary preliminaries for readers from diverse backgrounds to understand our paper, as well as to establish important notations for the subsequent method section. When drafting the paper, we made a deliberate effort to keep this section concise, limiting it to less than one page. That said, we recognize that the later sections may come across as dense due to the large volume of results and messages we aim to present. To address this, we will try to further shorten Sections 1-3. Additionally, if the reviewer could specify which key ideas are particularly challenging to follow, we would be more than happy to improve the corresponding explanations.

---

> ### Author Response · Authors · 2024-11-27
> **Response to Reviewer QTE1 4**
>
> **Q1:** Why is the full model performance significantly higher in this paper compared to the NAEE paper?
>
> -  Our evaluation is based on OpenCompass (source: https://github.com/open-compass/opencompass) while the original work of NAEE conducts evaluation using LM-Evaluation-Harness (source: https://github.com/EleutherAI/lm-evaluation-harness/tree/2a47159caff00135b026f724ace2a2011f3c7621).
> The difference in number is due to the different evaluation settings, including different prompt templates and answer-matching metrics. We did not use NAEE's evaluation approach as the NAEE paper appears during our study and experiments of EEP.
> However, we want to mention that, in our experiments, all methods are evaluated using OpenCompass, so the comparison is fair.  We adopt the default settings in OpenCompass to ensure less intervention in the evaluation.
>
> **Q4:** How is the performance of the proposed method compared to dynamic MoE methods?
>
> - From what we understand, dynamic MoE (e.g., [1]) refers to the family of MoE models where the number of activated experts is dynamically determined for each token. (Please let us know if you refer to other works.) In contrast, EEP is an MoE expert pruning method. These two approaches are orthogonal in nature and can complement each other rather than being competing methods.
>
>   - For example, our first use case focuses on reducing the total number of experts, making it orthogonal to dynamic MoE approaches. For example, one can apply EEP to dynamic MoE architectures so as to leverage the benefits of improved routing mechanisms while simultaneously achieving reduced parameter sizes.
>
>   -  For our second use case, as many dynamic MoE works set hyperparameters to control the sparsity of activated expert [1], which is similar to the K in the original top-K routing, EEP can also be used to achieve lower sparsity while maintaining the performance.
>
> - In summary, we think dynamic MoE is another type of work that can be combined with EEP, rather than being compared to it.
>
> [1] Huang et al., Harder Tasks Need More Experts: Dynamic Routing in MoE Models

---

> ### Author Response · Authors · 2024-12-01
> **Follow-up Comment**
>
> Thank you for your insightful review.
>
> Since your feedback has been crucial in refining our work, we have worked diligently in our rebuttal to address the concerns and questions raised. If you find that our response has satisfactorily resolved your concerns, we would kindly ask you to consider revisiting your score to better reflect your updated evaluation of the paper.
>
> Thank you again for your time and insightful review. If you have any further questions, we are very happy to discuss.

---

> ### Author Response · Authors · 2024-12-03
> **Follow-up Comment**
>
> Dear Reviewer QTE1,
>
> As the deadline of the discussion phase is approaching, we kindly ask you to review our response to see whether it can address your concerns. Following your suggestions, we have compared EEP with the fine-tuning method, presented detailed results of using less training data for both in-distribution and out-of-distribution pruning settings, and conducted experiments for OOD pruning with less validation datasets. If our response has addressed your concerns, would you mind reconsidering your score?
>
> Thank you once again for your effort and valuable feedback.

---

### Author Response · Authors · 2024-11-27
**General Response to all Reviewers**

We sincerely appreciate the reviewers' detailed and helpful suggestions on our work. We are excited to see that the reviewers think the paper is exceptionally well-written and organized (mRrF, oSj6, N5TL), the experiments are comprehensive (mRrF, N5TL), the approach is novel, well-justified, and interesting (mRrF, oSj6, N5TL), the analysis on router behavior is insightful (mRrF, oSj6), and the results are remarkable with significant improvements (mRrF, N5TL).

Following the reviewers' suggestions, we conducted additional 23 task-specific (in-distribution) search and evaluation experiments, 2 task-agnostic (out-of-distribution) search and more than 100 OOD evaluation experiments, as well as more computational cost profiling results, detailed in our rebuttals to each reviewer. We have also updated the revised manuscript (changes highlighted in red) based on reviewers' feedback.

Here, we want to highlight several points about our work.

- Firstly, we want to clarify a common misunderstanding of our work that *the search cost of EEP is high*.

    -  Although we did not focus on optimizing the search cost when designing EEP, **we have shown in our original submission that EEP's search cost can be easily reduced by using smaller training sets** (Table 15). To further support this, we added more experiments showing EEP's performance and search cost under 200, 100, and 50 data points (see the rebuttal and revision for details). The results show that **EEP can complete the pruning process in less than 20 minutes while achieving better performance than the full model and the baselines**. This computation time is comparable to the baseline method NAEE.

    -  Additionally, our computation time for EEP includes both the GPU time (model inference) and CPU time (model merging, which is the main overhead for small dataset size like 50). This makes the comparison somewhat unfair to EEP since GPU time is typically more expensive than CPU time. With further optimizations, such as performing model merging on GPUs, we expect EEP's computation time to be even faster.

- Secondly, we want to re-emphasize the key features of EEP that set it apart from baseline methods:

    - **Improving inference efficiency**: Through extensive experiments, we demonstrate that **EEP significantly improves the inference efficiency of MoE models by reducing both GPU memory consumption (by pruning the total number of experts) and inference latency (by pruning the number of active experts)**. Since deployment computational costs are often much higher than training costs (e.g., see [this report](https://www.datacenterdynamics.com/en/news/openai-training-and-inference-costs-could-reach-7bn-for-2024-ai-startup-set-to-lose-5bn-report/)), these benefits provided by EEP have substantial practical implications.

    - **Gradient-free:** Since EEP is a gradient-free method, **EEP requires much less GPU memory than gradient-based methods** (e.g., fine-tuning), and **it can run on devices available only for inference, expanding its applicability to more use cases.**

    - **Robustness in both in-distribution and out-of-distribution settings**: Extensive experiments demonstrate that EEP performs well regardless of whether the search and test datasets are the same or different. We notice that the reviewers are more interested in the out-of-distribution settings. But we want to mention that the **in-distribution (i.e., task- or domain-specific) setting where EEP outperforms the baselines significantly also demonstrates important applications**. Task-specific LLMs fine-tuned from general capability LLMs, such as Code LLama, BioMistral, and Llama Guard, are widely used in many applications. Recent papers [1] also argue that composing different task-specific models/modules rather than using one general-purpose model and expecting it to perform well on all tasks might be better in terms of scalability, efficiency, and so on. **Using EEP to adapt a general-purpose MoE model towards one specific downstream task while making it more efficient aligns with these trends.**

We hope that our rebuttal and revision effectively address the reviewers' concerns. If there are any follow-up questions, we would be glad to discuss them further.

[1] Xiao, Chaojun, et al. "Configurable foundation models: Building llms from a modular perspective." arXiv preprint arXiv:2409.02877 (2024).

---

### Meta-Review · Area_Chair_CFed · 2024-12-18

**Metareview:**

The deployment of Sparse Mixture-of-Experts models remains challenging due to their large parameter counts and GPU memory demands. To address this, this paper introduces Efficient Expert Pruning, a gradient-free evolutionary strategy that prunes and merges experts in SMoE models based solely on model inference, reducing parameters and memory usage while maintaining or improving performance. Experiments on multiple datasets and models, including Mixtral and Qwen, show that EEP achieves sparsity and inference speedups, outperforming baselines and enabling more efficient SMoE deployment.

Three reviewers have slightly positive feedback and most reviewers acknowledge the paper presentation & methods' efficacy, especially impressed by the high sparsity of compressing Mixtral MoE. However, it is well-known that Mixtral MoE is highly redundant, also for Qwen MoE. My major concern is the generalization ability of the proposed approach to other types of MoEs which have less redundancy due to improved optimization and designs. For example, deepseek and modularformer. Also, the discussions with related MoE merging papers are encouraged to be included. Overall, we recommend a rejection.

**Additional Comments On Reviewer Discussion:**

During the rebuttal period, the authors made a great rebuttal and put in lots of effort. We believe that incorporating all discussions will strengthen future revisions.

---

### Decision · Program_Chairs · 2025-01-22

Reject